# Transient fertilization of a post-Sturtian Snowball ocean margin with dissolved phosphate by clay minerals

Ernest Chi Fru [1] ✉, Jalila Al Bahri[1], Christophe Brosson[1], Olabode Bankole[2], Jérémie Aubineau [3], Abderrazzak El Albani[2], Alexandra Nederbragt[1], Anthony Oldroyd[1], Alasdair Skelton [4], Linda Lowhagen[4], David Webster[4], Wilson Y. Fantong[5], Benjamin J. W. Mills [6], Lewis J. Alcott[6], Kurt O. Konhauser [7] & Timothy W. Lyons[8]

Marine sedimentary rocks deposited across the Neoproterozoic Cryogenian Snowball interval, ~720-635 million years ago, suggest that post-Snowball fertilization of shallow continental margin seawater with phosphorus accelerated marine primary productivity, ocean-atmosphere oxygenation, and ultimately the rise of animals. However, the mechanisms that sourced and delivered bioavailable phosphate from land to the ocean are not fully understood. Here we demonstrate a causal relationship between clay mineral production by the melting Sturtian Snowball ice sheets and a short-lived increase in seawater phosphate bioavailability by at least 20-fold and oxygenation of an immediate post-Sturtian Snowball ocean margin. Bulk primary sediment inputs and inferred dissolved seawater phosphate dynamics point to a relatively low marine phosphate inventory that limited marine primary productivity and seawater oxygenation before the Sturtian glaciation, and again in the later stages of the succeeding interglacial greenhouse interval.

The secular rise of phosphorus (P) in Cryogenian marine sediments has been linked to post-Snowball Earth deglaciation resulting in enhanced phosphate ($PO_4^{3-}$) supply to seawater, greater marine primary production, ocean-atmosphere oxygenation, and ultimately the evolution of metazoans[1-3]. The inference of higher dissolved seawater $PO_4^{3-}$ concentrations across the Neoproterozoic glaciations comes from Fe-rich deposits and shales containing significantly higher P concentrations relative to older sediments of similar depositional settings[1,2]. However, except for a broad association to mechanical weathering caused by erosional action of end-Snowball melting ice sheets, it

remains unclear how $PO_4^{3-}$ was sourced from the continents and transported in quantitatively dissolvable forms to seawater to enable pervasive global ocean-atmosphere oxygenation.

In the modern world, most $PO_4^{3-}$ is transferred from land to the oceans by riverine-transported clay and metal oxide particles rather than in solution[4-6]. This view is consistent with the suggestion that detrital clay minerals associated with more acidic rivers during the Great Oxygenation Event (GOE) played a significant role in conveying $PO_4^{3-}$ from land to Paleoproterozoic seawater[5]. At times of less acid generation on land, highly reactive iron oxyhydroxides (Fe-ox$_{HR}$)

[1]College of Physical and Engineering Sciences, School of Earth and Environmental Sciences, Centre for Geobiology and Geochemistry, Cardiff University, Cardiff CF10 3AT Wales, UK. [2]Université de Poitiers UMR 7285-CNRS, Institut de Chimie des Milieux et Matériaux de Poitiers - 5, rue Albert Turpin (Bât B35), 86073 Poitiers, cedex, France. [3]Géosciences Environnement Toulouse, CNRS UMR 5563 (CNRS/UPS/IRD/CNES), Université de Toulouse, Observatoire Midi-Pyrénées, Toulouse, France. [4]Department of Geological Sciences, Stockholm University, 106 91 Stockholm, Sweden. [5]Institute of Geological and Mining Research (IRGM), Box 4110, Yaoundé, Cameroon. [6]School of Earth and Environment, University of Leeds, Leeds LS2 9JT, UK. [7]Department of Earth and Atmospheric Sciences, University of Alberta, Edmonton, Alberta T6G 2E3, Canada. [8]Department of Earth and Planetary Sciences, University of California, Riverside, CA 92521, USA. ✉e-mail: ChiFruE@Cardiff.ac.uk

instead served as efficient particulate sorbents for dissolved phosphate[5]. As a corollary, we hypothesize that extensive weathering of continental landmasses by melting ice sheets at the end of the Cryogenian Snowball Earth glaciations[7] drove up production, transport, and supply of fine-grained detrital clays and Fe-ox$_{HR}$ to the oceans, which served as vectors of marginal seawater enrichment with $PO_4^{3-}$. Indeed, melting of the modern Greenland ice sheet shows that up to 97% $PO_4^{3-}$ exported by glacial meltwater is associated with suspended sediment particles[8]. This relationship is predictable, given that clay minerals, which tend to form rapidly upon the retreat of glaciers with climate warming[9], together with Fe-ox$_{HR}$, possess large surface-to-volume ratios. Together, they contain positive charged surfaces at the pH range of rivers that make them efficient adsorbents of dissolved $PO_4^{3-}$ anions[4-6,10-17].

To assess the potential of clay minerals and Fe-ox$_{HR}$ minerals as vectors of dissolved $PO_4^{3-}$ to post-Snowball seawater, we collected marine sediments exposed on the Isles of Islay and the Garvellachs in the Dalradian Supergroup, Scotland, spanning pre-glacial, deglacial, and post-glacial phases of the Neoproterozoic Sturtian Snowball glaciation[7] (Fig. 1). The underlying pre-Snowball Tonian Lossit Limestone Formation (LLF) in the >4-km-thick Appin Group[18-23] is characterized by organic-rich mudstones, sandstones, carbonate, and shallow seawater stromatolites. The glaciogenic tillite cap of the LLF passes conformably[21] into the overlying ~1.1 km-thick Port Askaig Tillite Formation (PATF), correlated worldwide to Sturtian Snowball deposits based on geochronology, lithostratigraphy, and chemostratigraphy[18-23]. On the Garvellachs Island, the PATF appears as

twelve discrete diamictite units containing dolostone clasts overlain by the ~40-m-thick Great Breccia, comprised of deformed sediment bedrock rafts transported by icebergs from land to the ocean margin, and the 29 to 40-m-thick disrupted beds dominated by Fe-rich siltstones and dolostone concretions. Up-section, ~30 diamictite layers with interlayered sandstone units grade upwards into transported granitic clasts, which are in turn conformably overlain by the post-Snowball Bonahaven Dolomite Formation[18,19,21] (BDF). The BDF consists of ~300-m-thick fine-grained clastic rocks composed of siltstones and mudstones, with occasional stromatolitic dolostone and evaporitic deposits indicating deposition in arid greenhouse climate times[18-23].

In this study, we use a combination of geochemical techniques to demonstrate a causal relationship between continental erosion and the supply of dissolved $PO_4^{3-}$ to continental margin waters and associated temporal controls on seawater oxygenation dynamics across the Sturtian Snowball glaciation deposits of the Dalradian Supergroup.

## Results and discussion
### Bulk facies chemostratigraphy and mineralogy
Careful collection of outcrop samples took place in the field to avoid weathered and metamorphosed lithologies (Table S1). Comparative chemostratigraphic correlation of bulk facies geochemistry and $\delta^{13}C_{carbonate}$ and $\delta^{18}O_{carbonate}$ compositions (Figs. 2 & S1-S2) with previous studies of the same section[18-24], together with sediment mineralogy (Fig. 3 & Table S2), was used to evaluate data quality and to account for potential post-depositional alteration of primary geochemical composition. In addition, cross plots of Fe/Ti vs.

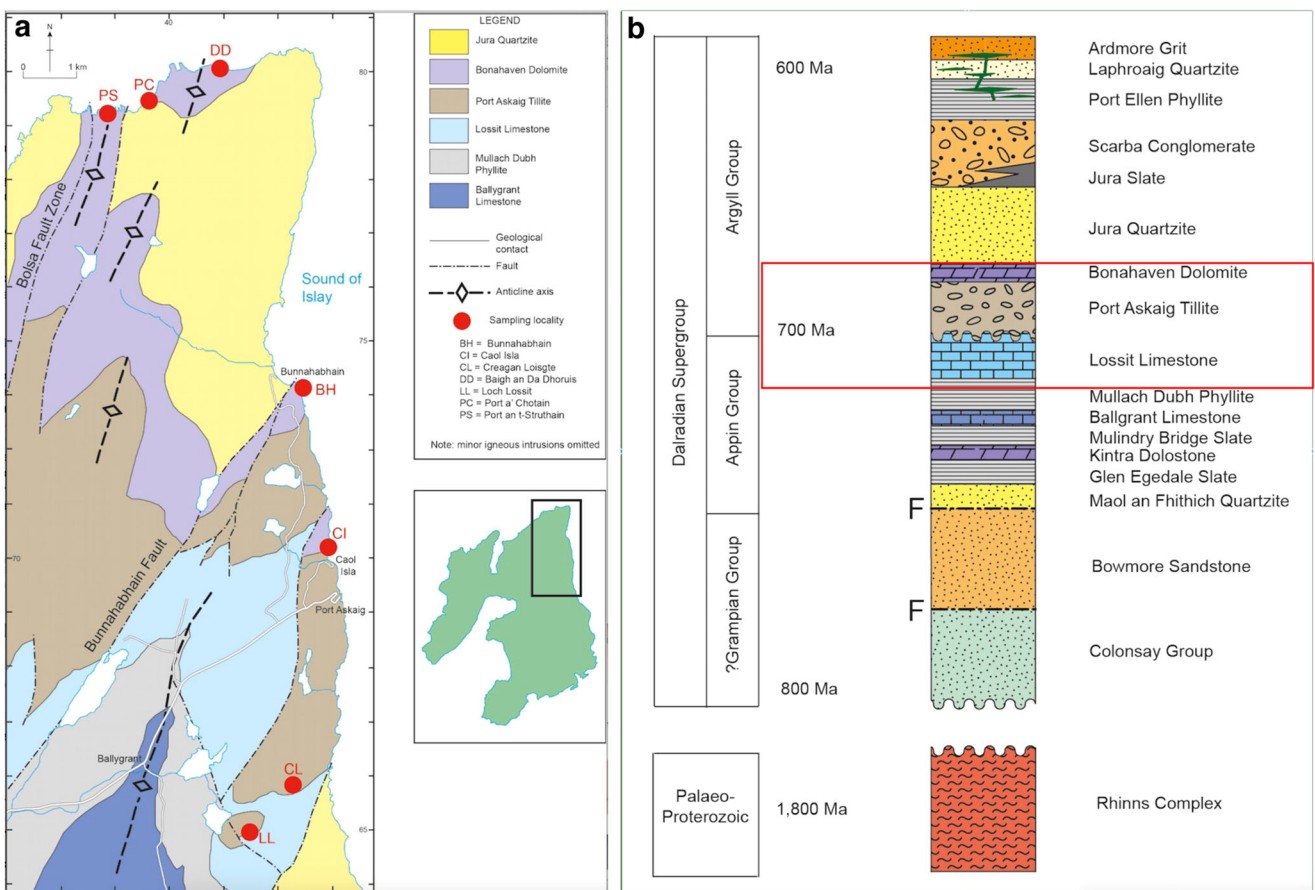

**Fig. 1 | Study locality, lithology, and stratigraphy. a** Geological map and lithology of the sampled sections. The black rectangular box in the green map, enlarged to scale to the left, displays the spatial distribution and location of sampled outcrops.

**b** Stratigraphy and ages within the Dalradian Supergroup of sampled outcrops boxed in red. LLF Lossit Limestone Formation, PATF Port Askaig Tillite Formation, BDF Bonahaven Dolomite Formation.

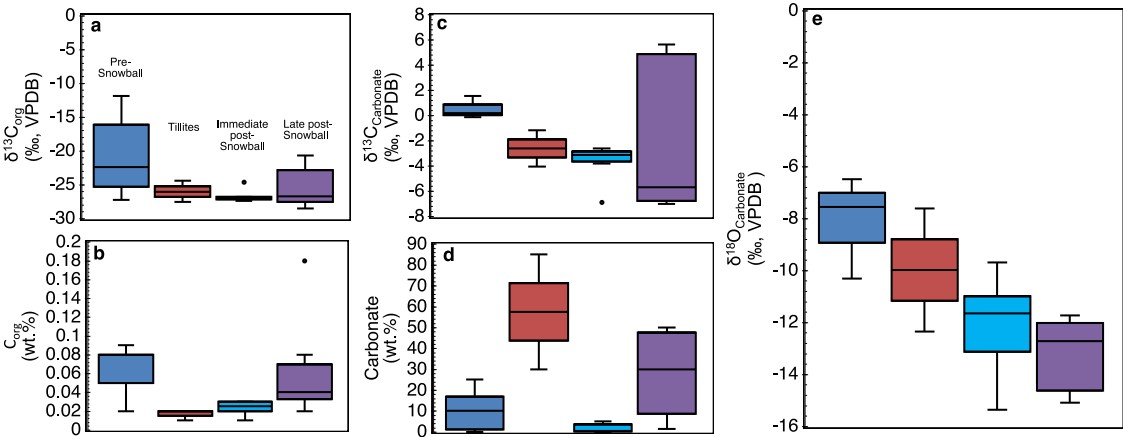

**Fig. 2 | Box and whisker plots showing lithostratigraphic carbon systematics for representative samples. a** $\delta^{13}C_{org}$ distribution. **b** Organic carbon ($C_{org}$) concentration. **c** $\delta^{13}C_{carbonate}$ distribution. **d** Carbonate concentration. **e** $\delta^{18}O_{carbonates}$ distribution. Centre line = median value; whiskers = minimum and maximum values; dots = outliers; box limits = lower and upper quartiles.

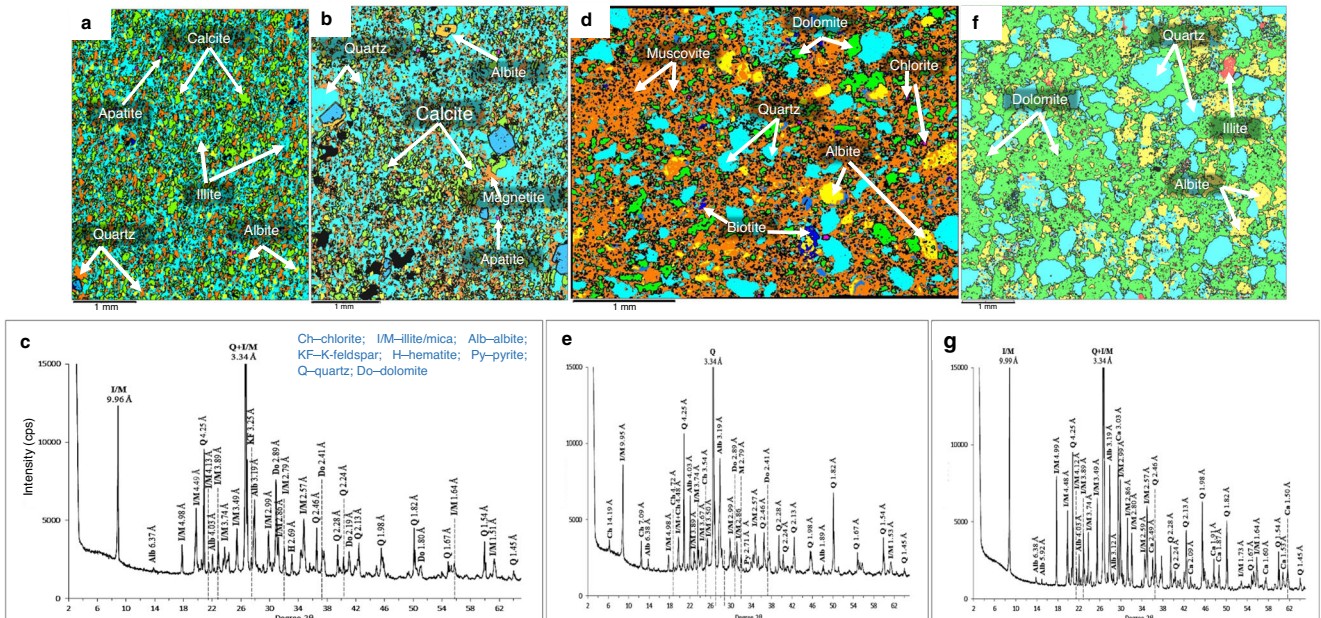

**Fig. 3 | Examples of Scanning Electron Microscopy-Energy Dispersive Spectroscopy (SEM-EDS) maps and bulk XRD mineralogical patterns for representative samples. a**, **b** SEM-EDS mineral maps for immediate post-Snowball, composed of fine-grain siliciclastic sediments. **c** Corresponding immediate post-Snowball whole rock XRD mineral diffactogram. **d** Representative SEM-EDS mineral map for Post Askaig tillite showing coarse siliciclastic grains dominated by fine orange muscovite phyllosilicate (sheet silicate) grains. **e** Corresponding XRD mineral diffactogram for the Port Askaig Tillites. **f** An example of an SEM-EDS mineral map for a pre-Snowball sample from the LLF, showing coarse siliciclastic grains, but with much lower phyllosilicate silicate content compared to the tillites. **g** Corresponding immediate pre-Snowball whole rock XRD mineral diffactogram.

Al/(Al+Fe+Mn) (ref. 25) were used to constrain the potential influence of hydrothermal fluids (Fig. S3). Our $\delta^{13}C_{carbonate}$ and $\delta^{18}O_{carbonate}$ values fall within the range reported in previous studies for the pre-glacial, glacial and post-glacial lithologies[18,19,21–23], which have been correlated by geochronology and chemo/lithostratigraphy to the Cryogenian-Sturtian interval in Australia, Mongolia, Siberia, China, Oman, Namibia and Canada[7,18,19]. The pre-/late-Snowball samples are marked by higher $\delta^{13}C_{org}$, $C_{org}$ and $\delta^{13}C_{carbonate}$ excursions, compared to persistently lower tillite and immediate post-Snowball values (Fig. 2a–c). Although characterized by lower carbonate content relative to concentrations in tillite samples analyzed for C isotope distribution (Fig. 2d), X-ray diffraction (XRD) and thin-sectioned Scanning Electron Microscopy-Energy Dispersive Spectroscopy (SEM-EDS) data are consistent with reported higher and more homogenous carbonate enrichment in the pre-/late-Snowball facies[18,19,21,22] (Fig. 3 & Table S2). Associated $\delta^{18}O_{carbonate}$ trends generally decline up sequence (Fig. 2e), while the absence of $\delta^{13}C$ and $\delta^{18}O$ correlation suggests negligible diagenetic alteration of primary $\delta^{13}C_{carbonate}$ signal (Fig. S2).

Lithologies affected by greenschist metamorphism, confined to the margins of metabasaltic sills[24], were excluded. Consistent with this sampling strategy, XRD and thin-sectioned SEM-EDS mineralogical analysis, revealed a major dominance of no to low-grade metamorphic clay minerals (Fig. 3 & Table S2). Typical clay minerals like kaolinite that are often associated with strong terrestrial weathering by chemical processes are rare in the samples, while muscovite and illite clay minerals believed to be most abundant in the modern

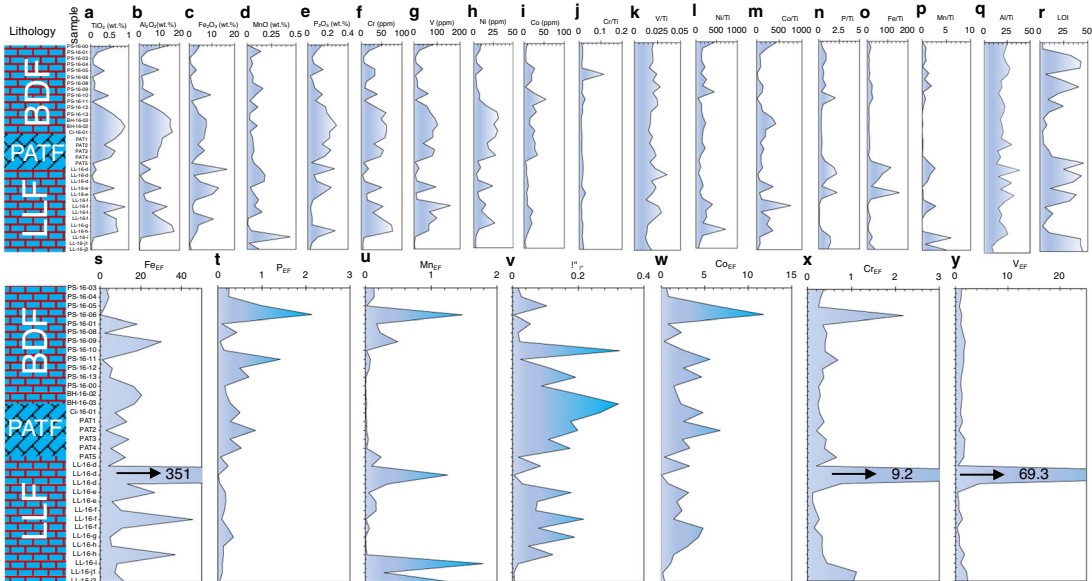

**Fig. 4 | Major and trace element distribution and lost on ignition (LOI) trends across the sampled sequence. a-e** Major element distribution. **f-i** Trace metal distribution. **j-q** Major and trace metal composition normalized to TiO₂. **r** LOI trend. **s-y** Trace element enrichment factors (EF), calculated as $EF_X = [(X/Al)_{sample}/(X/Al)_{UCC}]$, where X = concentration of element of interest according to Tribovillard and others (ref. 73). UCC, upper continental crust reference values according to Rudnick and Gao (ref. 74). LLF, Lossit Limestone Formation. PATF, Port Askaig Tillites. PAT, Port Askaig Tillites, where PAT1 and PAT2 represent tillites from the Port Askaig type location on Islay and PAT3 to PAT5 the LL-16-a sample at the top of the LLF that conformably underlines the Port Askaig tillites. BDF, Bonahaven Dolomite Formation. Major and trace element enrichment patterns.

upper continental crust (UCC)[17], occur more commonly across the lithofacies (Fig. 3 & Table S2). Chlorite, which also tends to be enriched in the UCC, was detected mainly in the deglaciating tillites, while the presence of albite (a tectosilicate mineral) in all samples regardless of lithology (Fig. 3), suggests an association with the erosion of crystalline bedrocks[19].

Overall, the SEM-EDS mineral maps together with XRD mineralogical analysis, show a variable prevalence of siliciclastic detrital material in all lithologies, with a dominance of sheet silicates and quartz, even in carbonate-rich facies. These findings agree with past observations[18–23], evidenced here by fine-grained siliciclastic particles in post-Snowball sediments and the prevalence of coarser quartz grains in tillite and pre-Snowball samples. The tillites are noted for hosting abundant detrital grains likely associated with illite, muscovite, albite, and, to a lesser extent, kaolinite (Fig. 3d-e & Table 2S). Illite and muscovite are quantitatively more abundant in the post-Snowball facies. Although detected in the pre-Snowball rocks, illite is not always present in readily quantifiable amounts. Typically, chemical weathering of most rock-forming silicates promotes secondary formation of clay minerals and Fe(III)- and Al-oxides, with an accumulation of residual quartz, heavy minerals, and sheet silicates like muscovite and biotite[26]. Sediments and soils originating from this process can predictably contain >50% sheet silicates and Fe oxyhydroxides by volume, depending on the nature of the source rocks and prevailing climatic conditions[17,26]. The variable occurrence of illite and muscovite across all sampled lithologies could be due to varying degrees of detrital supply or distinct local diagenetic pathways, while the largely low kaolinite content may be the result of possible transformation because kaolinite reacts with K-feldspar to form illite during burial diagenesis[17] and with alkaline fluids during metasomatism. Although chlorite can also form as a low-grade metamorphic mineral, with further transformation to muscovite at higher temperatures, the mineralogy of the facies is consistent with their suggested low metamorphic grade and observed preservation of primary features[18–23]. Further, the notable prevalence of quartz and sheet silicates (mainly clay minerals), including co-occurrence with tectosilicate albite, hint at the dominant

supply of clastic debris to the primary sediments, derived from physical erosion of intermediate-to-felsic continental rocks (Fig. S4).

## Bulk sediment geochemistry

A wide range of geochemical changes emerge along the studied lithostratigraphic section (Fig. 4 & Table S3), distinguished by persistently elevated bulk sediment P₂O₅, TiO₂, Al₂O₅, Fe₂O₃, Cr, V, and Ni concentrations in the PATF and the lower BDF, with much lower and more variable concentrations recorded in the PS-16 section of the Port an t-Struthain rocks in the upper BDF (Fig. 4a-c & f-h). Notable Al₂O₅ and TiO₂ peaks, especially in the PATF and lower BDF, agree with the prevalence of various mineral phases pointing to substantial detrital contribution to sediment accumulation across the sequence. To test this idea, elemental normalization to Ti—a detrital indicator of authigenic enrichment[27,28]—shows subtle enrichment consisting of two broad patterns across the stratigraphic profile (Fig. 4j-p). This relationship is highlighted by two observations. First, a reasonable stratigraphic correlation exists between P/Ti and Fe/Ti and to some degree with Mn/Ti ratios (Fig. 4n-p), although generally, MnO does not show significant lithostratigraphic variations (Fig. 4d). Second, a broad minimal correlation is observed for Cr/Ti, V/Ti, Ni/Ti, and Co/Ti (Fig. 4j-m), suggesting that their accumulation was co-regulated by the same sedimentary processes.

Interestingly, Cr/Ti is anomalous in the sense that in the lower BDF, when the other elements display multiple peaks, Cr/Ti distribution remains uniform, but becomes noticeable when enrichment in most elements is muted in the upper BDF Port an t-Struthain section (i.e., the PS-16 samples). To add to these observations, the near static Al₂O₃ to TiO₂ ratios (Fig. 4q) are interpreted to either reflect stable detrital contribution and/or potential long-term sediment accumulation from a similar UCC rock source.

Comparable Fe₂O₃ to TiO₂ and P₂O₅ to TiO₂ ratios, together with Loss on Ignition (LOI) being inversely correlated with P₂O₅ and Fe₂O₃ content throughout the sampled section (Fig. 4r & Fig. S5), suggest bulk sediment P₂O₅ and Fe₂O₃ are most likely preserved in inorganic instead of organic and carbonate mineral phases. Where LOI values are elevated, they presumably predict higher combustible sedimentary

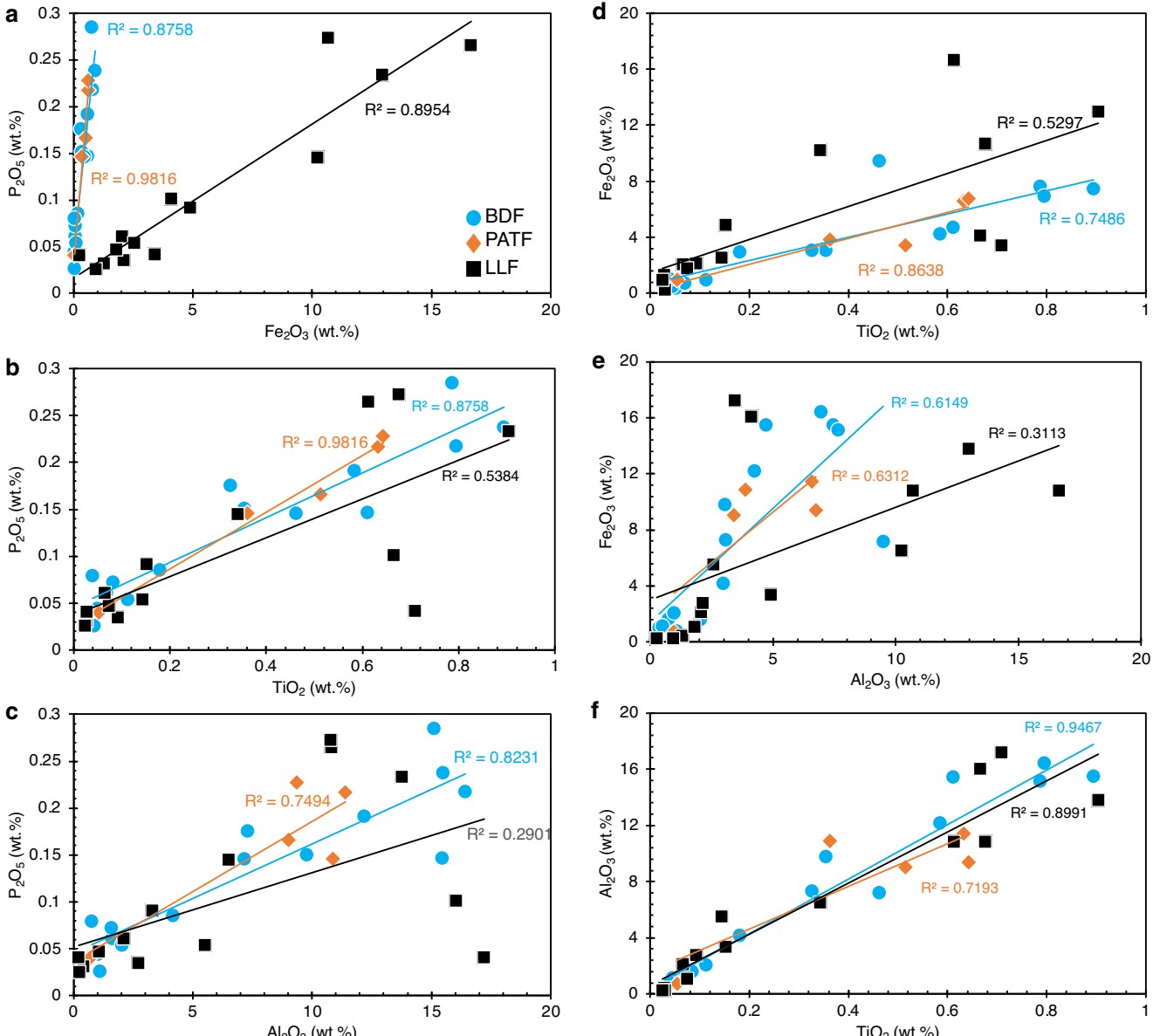

**Fig. 5 | Major trace elements cross plots for pre-Snowball (LLF), Tillites (PATF) and post-Snowball (BDF). a** $P_2O_5$ versus $Fe_2O_3$. **b** $P_2O_5$ versus $TiO_2$. **c** $P_2O_5$ versus $Al_2O_3$. **d** $Fe_2O_3$ versus $TiO_2$. **e** $Fe_2O_3$ versus $Al_2O_3$. **f** $Al_2O_3$ versus $TiO_2$. LLF, Lossit Limestone Formation. PATF, Port Askaig Tillites. BDF, Bonahaven Dolomite Formation.

organic and/or carbonate content, while lower values indicate the contrary, due to higher susceptibility of carbonates and organic matter to complete combustion to $CO_2$ under elevated temperatures[29]. Indeed, the lowest LOI values are recorded in the carbonate-poor PATF and lower BDF section where sustained peak $P_2O_5$ and $Fe_2O_3$ concentrations are found, compared to the minor and more variable $P_2O_5$ content in the carbonate-rich LLF and upper BDF PS-16 facies (Figs. 4c, e & 3r). Higher LOI values that tend to relate inversely with $P_2O_5$ and $Fe_2O_3$ concentrations across the section (Fig. S5), are unique to the carbonate-rich LLF and PS-16 upper BDF samples. Measured trace metals are generally enriched across the sequence relative to UCC values (Fig. 4s-y). Notably, significant co-enrichment of P is also observed in the tillites and upwards relative to the UCC, compared to pre-Snowball times (Fig. 4t). Despite their overall similar redox properties, Mn enrichment patterns relative to the UCC are dissimilar to those of Fe, particularly in some tillite and immediate post-Snowball samples (Fig. 4s & u), pointing perhaps to different enrichment

pathways or the known differences in sensitivity to reduction. Chromium shows two anomalous enrichment peaks relative to UCC, one in the LLF that is also seen for V and Fe and a second corresponding to samples shown above to have high Cr/Ti ratios in the upper BDF (Fig. 4j, s, x & y).

The generally low $C_{org}$ levels across the sampled sequence (Fig. 2b) are considered to be of little importance to sedimentary P and Fe preservation. Positive correlations are revealed for $P_2O_5$ and $Fe_2O_3$ and poorly mobile $TiO_2$ and $Al_2O_3$ across lithostratigraphy, although they are much weaker with respect to $Al_2O_3$ for the pre-Snowball Earth LLF samples (Fig. 5a-c). A similar relation appears for $Fe_2O_3$ versus $TiO_2$ and $Al_2O_3$ plots, but like $P_2O_5$ the correlation is unsupported for $Al_2O_3$ in the LLF section (Fig. 5d-e). When considered in light of the strong positive correlation between $Al_2O_3$ and $TiO_2$ (Fig. 5f) and $Fe_2O_3$ and $P_2O_5$ (Fig. 5a) in all three lithological sections, the observations suggest some level of decoupling of pre-Snowball $P_2O_5$ and $Fe_2O_3$ enrichment pathways with respect to $Al_2O_3$ sources (Fig. 5d-f). The strong

correlations between $TiO_2$, $P_2O_5$ and $Fe_2O_3$ for tillites and post-Snowball facies may consequently be the result of sediment source homogenization by the indiscriminate erosional action of melting ice sheets. Regardless, these observations provide further evidence for across sequence $P_2O_5$ and $Fe_2O_3$ enrichment by detritus.

The low $C_{org}$ and negative $\delta^{13}C_{carbonate}$ distribution in the facies (Fig. 2 & S1-S2) suggest microbial oxidation of $C_{org}$ likely prevailed at the sediment-water interface, which would have promoted the release of organic-bound P into sea and sediment pore water, influencing dissolved aqueous P content and eventual incorporation into various sedimentary minerals. This would have been particularly relevant in the immediate post-Snowball interval where apatite formation became more pronounced, compared to the rest of the section (Fig. 3 & Table S2). It is also possible that prevailing environmental conditions influenced ongoing microbial transformation of P-rich $C_{org}$ and Fe-$ox_{HR}$ minerals to trigger enough diagenetic sediment porewater P supersaturation[30]. For example, the geochemical data suggest that bulk primary sediment P was predominantly associated with non-calcium bearing minerals such as Fe-$ox_{HR}$ and unreactive detrital silicate phases (Figs. S6-S9), which would have starved primary sediments of dissolved P. Except for some immediate post-Snowball samples, across-sequence CaO and $P_2O_5$ inverse correlations, being up to 79% for the tillites (Fig. S6a-b), corroborate limited potential P preservation in non-calcium bearing mineral phases. As discussed above, this view is supported by low LOI, low-carbonate facies associated with bulk high $P_2O_5/Fe_2O_3$ ratios and high LOI, high-carbonate lithologies with bulk low-$P_2O_5/Fe_2O_3$ ratios (Fig. S5). Overall decreasing $P_2O_5$ concentration coincides with $C_{org}$ content that increases from post-Snowball to pre-Snowball facies (Fig. S6c), signalling either possible diagenetic P enrichment or loss in primary sediments through microbial oxidation of organic-rich P biomass originating from the water column. Indeed, our lowest $\delta^{13}C_{org}$ and $C_{org}$ values in the tillites and immediate post-Snowball interval (Fig. 2a-b) coincide with $P_2O_5$ enrichment compared to the low $P_2O_5$ pre-Snowball samples (Fig. S6c-d). Based on these observations, microbial recycling of organic-rich P at the sediment-water interface could explain the increasing prevalence of apatite in the immediate post-Snowball interval. However, the data suggest that the P supersaturation required to spontaneously precipitate large amounts of apatite precursor phases and significant substitution in carbonate minerals[30,31] was limited, especially in the pre-Snowball interval. Instead, our results indicate that dominant P supply and enrichment in the primary sediments mostly reflect detrital loading with non-apatite minerals. Moreover, as we show below, the synthesis of Ca-rich P minerals was perhaps limited by abundant Fe-$ox_{HR}$ particles acting as efficient scavengers of dissolved $PO_4^{3-}$ at the sediment seawater interface.

Redox-sensitive Mn shows a weak negative correlation with $TiO_2$ and $Al_2O_3$ across all facies (Fig. S10), suggesting that Mn enrichment pathways are to some extent decoupled from those of $P_2O_5$ and $Fe_2O_3$, particularly given the considerable positive correlation between $P_2O_5$, $TiO_2$, and $Al_2O_3$ (Fig. 5). Except for the short-lived spike in Cr enrichment in the upper BDF relative to Ti, overall, our data do not indicate that significant oxidative/chemical weathering of terrestrial rocks was a predominant mechanism for $PO_4^{3-}$ delivery to seawater throughout the sampled section. This observation is consistent with independent findings pointing to primary sediment detrital inputs of continental provenance[18,20–23], reinforced by our stronger physical rather than chemical weathering indicators. This relationship is further strengthened by the association of average UCC profiles (Fig. S11) with the continental margin location of the sedimentary basin.

High Cr/Ti ratios recorded during the GOE were previously ascribed to mobilization of dissolved Cr from land to the ocean by acid rock drainage via the activity of acidophilic aerobic bacteria following oxygenation of the atmosphere[27]. A similar but previously unreported acidification event could explain the brief spike in Cr/Ti ratios in the

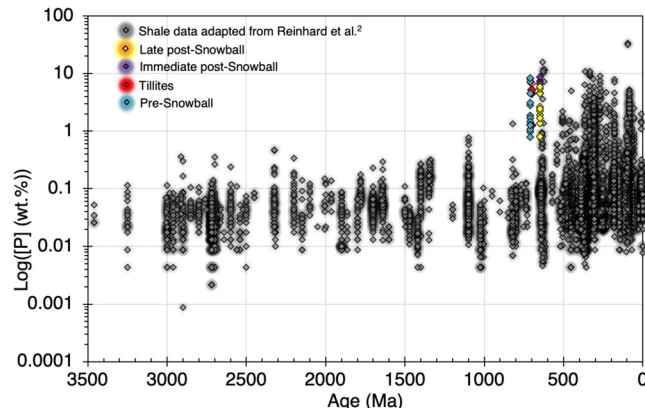

**Fig. 6** | Distribution of sample P concentration relative to fine grain marine siliciclastic sedimentary rock content through time[2].

upper BDF. However, a close look at the data indicates that total $P_2O_5$ and $Fe_2O_3$ contents immediately before, during and after the transient increase in Cr/Ti ratios remained low, suggesting that even if this event was triggered by acidic weathering of terrestrial rocks, it cannot explain the behaviour, source, and mechanism of $P_2O_5$ enrichment in our samples.

Further, the highly soluble properties of ferrous and ferric Fe under extreme acidic conditions characteristic of acid rock drainage[32] should translate to relative co-enrichment of Fe and Cr in the sediments with respect to poorly mobile Ti. This relationship is not seen in the studied section. Moreover, the broad expression of similar but conservative Fe/Ti and P/Ti profiles throughout the BDF and the tillite formation are consistent with reported low to moderate chemical weathering indices in the tillites[33]. Thus, the combined data better reflect terrestrial detritus contributing to high bulk $P_2O_5$ and $Fe_2O_3$ enrichment in the PATF and the lower BDF section, assuming continental erosion by melting ice sheets was the principal source of detrital inputs into the sediment pile. Moreover, our bulk P data are consistent with historical records showing a substantial spike in the P content of Cryogenian fine-grained marine siliciclastic rocks[2] (Fig. 6).

## Highly reactive Fe and dissolved P chemistry

In light of the above discussions, we pose the question of whether P was associated with Fe in the primary sediments and, if so, whether such a relationship can provide insights into seawater-dissolved P profiles. The distribution of leachable highly reactive Fe ($Fe_{HR}$) phases in the form of Fe-$ox_{HR}$ + pyrite Fe and poorly reactive sheet silicate Fe (Fig. 7a) in representative samples was therefore explored according to references[34–36] (see methods). Intriguingly, the Fe-$ox_{HR}$ phases decline by a factor of two when emerging from the Snowball ice sheet melting stage and transitioning into post-Snowball greenhouse state (Fig. 7b). The ratio of Fe-$ox_{HR}$ over total Fe content, suggest that Fe-$ox_{HR}$ constitutes only a small proportion of total Fe across the sequence, with distinct fluctuations that parallel low pre-Snowball $P_2O_5$, immediate post-Snowball high $P_2O_5$ and late post-Snowball low $P_2O_5$ intervals (Fig. 7c). A similar pattern is noted when Fe-$ox_{HR}$ concentrations are compared to unreactive sheet silicate Fe content, with the data suggesting that the majority of Fe throughout the sampled stratigraphy is largely present as unreactive silicates. These data are consistent with the above observations pointing to significant detrital contribution to sediment Fe enrichment, as opposed to chemical sedimentation.

Bulk sediment P analysis, as shown by our results, provides important clues on the overall behaviour of P in marine waters[1,2,37,38]. However, bulk sediment P content is a poor differentiator of reactive and unreactive P composition and therefore is an unreliable indicator of P bioavailability. Instead, we consider the linear correlation between

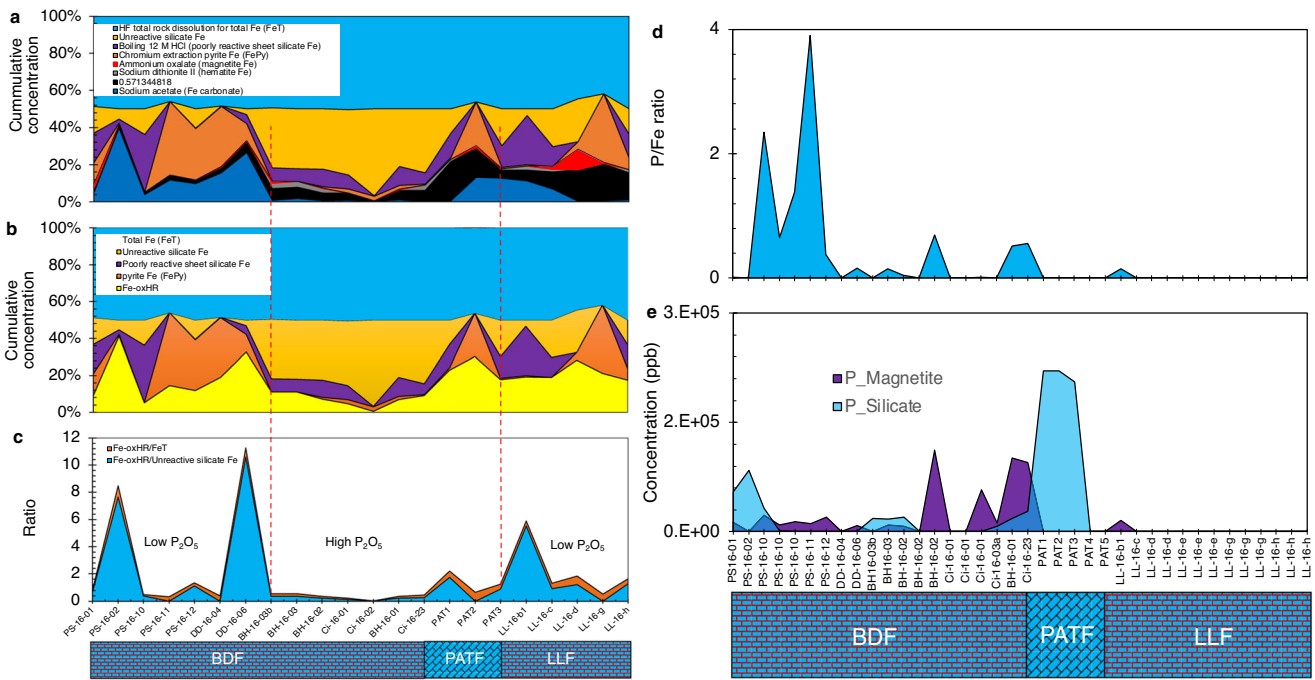

**Fig. 7 | The distribution of reactive, poorly reactive, and unreactive Fe mineral phases. a** Contribution of defined Fe mineral phases expressed as a percentage of total Fe content across sequence stratigraphy. **b** Contribution of highly reactive Fe oxyhydr(oxide) (Fe-ox$_{HR}$), poorly reactive sheet silicate Fe, unreactive sheet silicate Fe and pyrite Fe, expressed as a fraction of total Fe content across sequence stratigraphy. **c** Fe-ox$_{HR}$ to total Fe ratio (Fe$_T$) (Fe-ox$_{HR}$/Fe$_T$) and Fe-ox$_{HR}$/unreactive Fe ratio. **d** Representative putative magnetite- and sheet silicate-bound P distribution across sequence stratigraphy. **e** Representative putative magnetite-bound P/Fe ratios across sequence stratigraphy. LLF, Lossit Limestone Formation. PATF, Port Askaig Tillite Formation. PAT, Port Askaig Tillites, where PAT1 and PAT2 represent tillites from the Port Askaig type location on Islay and PAT3 the LL-16-a sample at the top of the LLF that conformably underlines the Port Askaig tillites. BDF, Bonahaven Dolomite Formation.

Fe-ox$_{HR}$-bound P and dissolved P (refs. [1],[37–39]). in modern marine systems as the best estimator of dissolved P behaviour during sediment deposition, following the recommendation of Thompson and others[40] (see methods). We found 0% of measurable leached P associated with the Fe carbonates, compared to ~9.1% for the combined goethite, akageneite and hematite phases (all post-Snowball); ~45.5% for magnetite (42.5% post-Snowball and 3.0% pre-Snowball); ~6.1% for hematite (all pre-Snowball); and 39.4% for the tillites and post-Snowball sheet silicates (Figs. S7 & S8).

The sheet silicates in the tillites and immediate post-Snowball, record the highest leachable P content. Overall, the distribution of P between the sheet silicates and the Fe-ox$_{HR}$ reservoir indicate that high P enrichment in association with sheet silicates at the terminal Snowball immediate post-Snowball greenhouse transition, is marked by redistribution of P to a putative magnetite sink (Fig. 7d). A corresponding ~twofold reduction in the size of the leachable Fe-ox$_{HR}$ inventory at this time would have reduced the potential removal efficiency of dissolved PO$_4^{3-}$ from seawater by an equally similar magnitude, since Fe-ox$_{HR}$ is otherwise a strong P sink[37–39]. This process alone would have enabled substantial build-up of dissolved PO$_4^{3-}$ in immediate post-Snowball seawater relative to the period before and after, both distinguished by a higher Fe-ox$_{HR}$ reservoir (Fig. 7b). Furthermore, the peaking of putative sheet silicate P in the tillites, which is hundreds of times above baseline detection concentrations of 5.47 ppb in the pre-Snowball interval, is consistent with rock flour production expected with the mechanical grinding action of melting ice sheets on the bedrock. This supposition is supported by SEM-EDS imaging showing elevated fine-grained sheet silicate clasts in the immediate post-Snowball samples (Fig. 3d).

We assume that the coincidental rise of appreciable P enrichment in immediate post-Snowball magnetite grains relative to the tillites interval, points to increasing immediate post-Snowball

co-precipitation of magnetite and dissolved PO$_4^{3-}$ out of seawater. A test of this hypothesis finds that peaks in magnetite-associated P/Fe ratios correlate with the appearance and persistence of magnetite-bound P at the tillite post-Snowball boundary and upwards (Fig. 7d-e). This up-section correlation between magnetite-bound P concentration and magnetite P/Fe ratios, suggests magnetite precipitation captures a snapshot of the potential dissolved PO$_4^{3-}$ content of the reservoir from which it formed. The distinctly variable but higher P/Fe values in the late low-P$_2$O$_5$ post-Snowball interval are best explained by emergent long-term moderate presence and persistence of magnetite-bound P in the facies (Fig. 7d). The mostly non-variant magnetite-bound P trend at this time, with the exception of a single spike, points to steady scavenging of PO$_4^{3-}$ by an increasingly scarce magnetite reservoir, instead of a rise in dissolved PO$_4^{3-}$ content.

The release of bound PO$_4^{3-}$ to seawater and/or sediment pore water by diagenetic dissolution of Fe-ox$_{HR}$ minerals could, however, account for the two times smaller Fe-ox$_{HR}$ budget in the immediate post-Snowball P$_2$O$_5$-rich rocks, compared to the P$_2$O$_5$-depleted sediments that bracket this interval. Yet, because diagenesis results in considerable loss of Fe-ox$_{HR}$-bound P, this interval should be accompanied by parallel lowering of sedimentary P$_2$O$_5$ and Fe-ox$_{HR}$-associated P content if this were the case. Instead, we observe co-increase in both bulk P$_2$O$_5$ and potential magnetite-bound P at this time. Further, the comparatively lower pre/late post-Snowball P$_2$O$_5$ and magnetite P reservoirs cannot be explained by reductive dissolution of Fe-ox$_{HR}$ since their Fe-ox$_{HR}$ inventories are about twice the size found in the P$_2$O$_5$-rich immediate post-Snowball section. In addition, the transition of bulk Fe/Al ratios from the higher values expected with enhanced authigenic Fe enrichment during the pre-Snowball time to lower detrital Fe/Al ratios averaging ~0.5 in the tillites and for most of the post-Snowball sediments, is consistent with negligible reductive Fe mobilisation by diagenetic microbial Fe-reduction[34,41,42]. Importantly,

the enrichment of P in the leachable immediate post-Snowball magnetite reservoir, which comes after a rise in associated sheet silicate-bound P, is best explained by transfer of sheet silicate-bound P to seawater, followed by chemical enrichment in the sediment pile by Fe-ox$_{HR}$ produced in the greenhouse interval.

Our estimates are conservative within reasonable error margins, considering that the designation of extractable Fe-ox$_{HR}$ phases in the defined Fe mineral pools can at times be misleading. For example, goethite was previously detected in the ammonium oxalate extract attributed to magnetite[43]. Our data are nonetheless consistent with SEM-EDS mineral imaging and bulk XRD mineralogical analysis, which failed to detect putative goethite-enriched phases as a major mineral constituent in the samples, suggesting an absence or low levels below detection. Instead, putative crystalline magnetite grains were observed in the immediate post-Snowball section, but rarely in the tillites and the pre- and post-Snowball intervals, interpreted as either an absence or very low concentrations (Fig. S9). This observation is consistent with potential pre-Snowball magnetite contributing only an estimated ~5.4% P to the total extractable P pool, compared to up to 37.8% for the post-Snowball time.

## Magnetite precipitation and seawater dissolved P behaviour

Magnetite, a common Fe mineral in Precambrian Fe-rich sedimentary rocks, often forms through a combination of biotic and abiotic processes involving ferrihydrite and green rust precursors[44–48]. In this light, green rust and ferrihydrite are thought to sequester and bury dissolved seawater trace nutrients in the sedimentary pile[49,50]. Particularly, the ability for both green rust and ferrihydrite to bind aqueous $PO_4^{3-}$ has been demonstrated experimentally[37–39,51–54], with green rust reported to possess a greater propensity to bind $PO_4^{3-}$ than ferric Fe[53]. It is thus reasonable to assume that the incorporation of P into magnetite crystals scaled proportionally with the concentration of primary seawater green rust and ferrihydrite magnetite precursors.

Nonetheless, although prograde metamorphism exerts a negligible effect on magnetite transformation, it can become significant in the presence of olivine[55,56]. Mineralogical appraisal of our samples by XRD and SEM-EDS mapping, however failed to identify olivine as an important mineral phase throughout the sequence, while the influence of hydrothermal fluids on Fe distribution appears to be insignificant (see Fig. S3). Moreover, a lack of covariation between carbonate Fe and magnetite Fe reservoirs across sequence stratigraphy, further suggests little or no thermal breakdown of Fe carbonates to magnetite by burial metamorphism[57]. Importantly, spontaneous binding of $PO_4^{3-}$ to magnetite occurs by physical adsorptive intermolecular attractive forces and by chemical transfer of electrons between magnetite and $PO_4^{3-}$ molecules, promoting an adsorption capacity of ~57.8 mg of $PO_4^{3-}$ per gram and resulting in a positive correlation between magnetite-bound and dissolved $PO_4^{3-}$ (ref. 13). This binding of dissolved $PO_4^{3-}$ by magnetite enhances magnetite's stability[12,13] and therefore enhancement of possible long-term preservation in sediments.

We can only speculate that the prevalence of putative magnetite grains, particularly in the immediate post-glacial P$_2$O$_5$-rich interval, compared to the low P$_2$O$_5$ intervals, reflects nascent post-glacial seawater conditions, probably related to the balance between seawater oxygen content and local Fe sedimentary mechanisms. For instance, modern turbid Arctic glacial outflows generally display high dissolved Fe content of up to 20 μM, which decreases to nanomolar concentrations at the shelf-fjord water mixing boundary[58]. This trend results from rapid flocculation of dissolved Fe at this interface and the binding of particle surfaces because of increasing salinity and pH, triggering the loss of up to 98% total dissolved Fe (refs. 58,59). Similar rapid precipitation of particulate Fe with increasing oxygenation at Snowball melt water-seawater interfaces would have accelerated the scavenging of dissolved P released by clay minerals along the gradients of rising salinity and pH in the continental margin waters[5,6].

Although we are unable to quantitatively distinguish the extent of Fe(II) oxidation at the end-Snowball post-Snowball transition, predominantly positive δ$^{56}$Fe$_{bulk}$ sedimentary values (Table S4) indicate that partial Fe(II) oxidation[60] was common throughout the deposition of the studied sequence. This process would have favoured preferential formation of magnetite as allowed by local thermodynamic conditions of redox, electrical conductivity, pH, and temperature[61,62]. For instance, magnetite precipitation and stability may have been favoured at the lower temperatures and pH conditions that accompanied the immediate post-glacial CO$_2$-rich world[61,62]. It is also possible that the reduction of ferrihydrite by biomass to magnetite could have been common[45,47]. However, the absence of strong negative δ$^{56}$Fe signatures across the studied profile is consistent with overall negligible diagenetic production of magnetite and Fe carbonates via dissimilatory microbial reduction of Fe-ox$_{HR}$. Thus, considering the linear relationship existing between magnetite-bound and aqueous $PO_4^{3-}$ (ref. 12), $PO_4^{3-}$ bioavailability in the immediate post-Snowball waters potentially increased by at least 20-fold compared to pre-Snowball conditions. Unlike the immediate post-Snowball interval, putative immediate pre-Snowball magnetite-bound P remained below the instrument detection limit of ~5 ppb, with the data indicating that unreactive silicate-bound P dominated bulk P burial before and during the Sturtian Snowball (Fig. S7).

## Implications for seawater redox

Carbonate-rich samples with <5% Fe content tend to produce anomalously high Fe$_{HR}$/Fe$_T$ ratios[34]. The reconstructed Fe-based redox trends were therefore evaluated for deviations as a function of Fe and carbonate content. The absence of significant differences regardless of total Fe and carbonate content resulting in consistent trends along sequence stratigraphy, was taken to imply a negligible impact of carbonate content on the reliability of the Fe-based redox proxy. We further interpret the general lack of significant deviations in the Fe-based redox proxy profiles as a reflection of the significant siliciclastic and Fe-rich content of the studied lithologies[18–23], including those containing carbonates, as established by our geochemical analyses. The impact of sedimentation rates was determined to be negligible, considering that estimated values are four times lower than those expected to compromise the sensitivity of the Fe-based redox proxy[34] (see supplementary information).

Average Fe$_{HR}$/Fe$_T$ ratios are $0.55 \pm 0.21$, $0.16 \pm 0.04$, and $0.68 \pm 0.30$ for the pre-Snowball and tillites, the P$_2$O$_5$-rich immediate post-Snowball, and the P$_2$O$_5$-poor late post-Snowball intervals, respectively (Fig. 8a). Their corresponding average Fe$_{Py}$/Fe$_T$ ratios are $0.21 \pm 0.22$, $0.31 \pm 0.34$, and $0.42 \pm 0.31$, respectively (Fig. 8b). The Fe$_{py}$/Fe$_{HR}$ ratios of <0.8 suggest deposition of the entire sediment profile in mainly ferruginous-like conditions, although the Fe$_{HR}$/Fe$_T$ ratios of <0.22 in the immediate post-Snowball interval are consistent with oxygenated waters, while the >0.38 Fe$_{HR}$/Fe$_T$ ratios indicate anoxic bottom water depositional conditions for the pre-Snowball, tillite, and late post-Snowball samples[34–36]. Average Fe$_{HR}$/Fe$_T$ values for samples straddling the peak post-glacial P$_2$O$_5$ interlude correlate negatively with their associated magnetite-bound P values, implying rising magnetite-bound *P* values coincide with lower Fe$_{HR}$/Fe$_T$ ratios associated with oxic waters (Fig. 8c). No such correlation is observed for the pre-Snowball, tillites and late post-Snowball samples. These observations provide substantial evidence that the transitory rise in sedimentary P$_2$O$_5$ content in the immediate post-Snowball sediments was likely attended by transient seawater oxygenation and that magnetite indeed records dissolved seawater $PO_4^{3-}$ dynamics at this time.

The oxygenation trends are further highlighted by temporal Fe/Al profiles across the sequence[34,41,42,63] (Fig. 8d). In this regard, the >0.5 pre-Snowball Fe/Al ratios are generally consistent with higher sediment pyrite enrichments, compared to tillites samples and the immediate post-Snowball facies, which are depleted in pyrite

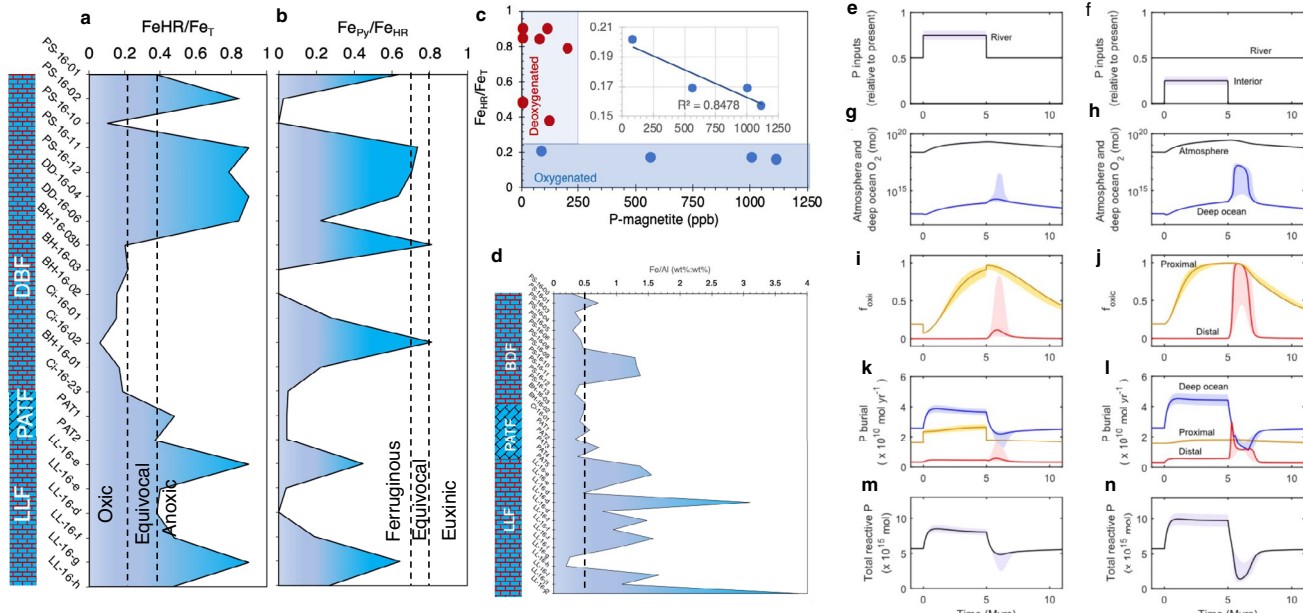

**Fig. 8 | Redox reconstruction. a** $Fe_{HR}/Fe_T$ ratios. **b** $Fe_{py}/Fe_T$ ratios. **c** Relationship between associated magnetite-bound P and Fe speciation. **d** Fe/Al trend for the studied section. **e** Numerical model with riverine P input increased by 40-60% relative to present day for 5 Myr. **f** Numerical model for atmosphere and deep ocean oxygen reservoirs for river input scenario. **g** Numerically modelled fraction of oxic seafloor in the proximal and distal shelf environments for river input scenario. **h** Numerical model when riverine P input is held constant relative to present day, while a further 40–60% of background P flux is added to ocean interior. **i** Numerical model for atmosphere and deep ocean oxygen reservoirs for ocean interior input scenario. **j** Numerically modelled fraction of oxic seafloor in the proximal and distal shelf environments for ocean interior input scenario. **k-l** Numerically modelled P burial fluxes in the deep ocean, proximal shelf, and distal shelf environments. **m-n** Numerically modelled total marine P concentration. Note that interior ocean P release results in a higher marine P inventory by avoiding rapid burial in the proximal zone. For full details of the model, see methods and references[63,64]. LLF, Lossit Limestone Formation. PATF, Port Askaig Tillite Formation. BDF, Bonahaven Dolomite Formation.

(Figs. 7b & 8b). Such high Fe/Al values have been linked to the deposition of anoxic sediments beneath modern sulphide-enriched seawater, and lower values characterize oxic settings with limited $C_{org}$ supply to sediments[41,42]. Due to persistent supply of continental detritus as shown above, together with overall sedimentation rates that do not compromise the $Fe_{HR}/Fe_T$ proxy as discussed in the supplementary information, the >0.5 Fe/Al ratios are best explained by syngenetic changes in anoxic pyrite precipitation. The <0.5 Fe/Al ratios, however, do not effectively delineate the pyrite-poor anoxic tillites from the apparently oxic immediate post-Snowball deposit (Fig. 8a), possibly due to increase input of Al-rich detritus during deposition of the tillites and immediate post-Snowball facies (Fig. 4b). Similarly, <0.5 Fe/Al ratios in the anoxic pyrite-poor late post-Snowball interval, as also suggested by the $Fe_{HR}/Fe_T$ redox proxy, are followed by a small >0.5 Fe/Al spike corresponding to anoxic conditions (Fig. 8a) and pyrite enrichment at the top of the section (Figs. 7b, 8b & 8d).

Furthermore, a numerical biogeochemical model[64,65] (see methods) replicates the observed temporal rise and fall in seawater $PO_4^{3-}$ content and the transient immediate-post Snowball oxygenation event (Fig. 8e-j). The model which considers abrupt large input of riverine $PO_4^{3-}$ to near continental margin waters and the deep ocean, suggests that the initial considerable introduction of $PO_4^{3-}$ to continental margin waters could have briefly triggered seawater deoxygenation due to eutrophication (Fig. 8i). We note that the early rise in sedimentary $P_2O_5$ content was not immediately accompanied by seawater oxygenation. Moreover, the model shows that rising dissolved seawater $PO_4^{3-}$ concentration would have promoted increased $PO_4^{3-}$ burial, with subsequent decline in supply resulting in seawater deoxygenation (Fig. 8k-n), similar to our observations.

Our data reveal bulk P content comparable to previously published concentrations for fine grain siliciclastic Cryogenian facies. These data suggest that Cryogenian continental seawater P bioavailability before, during, and after the Sturtian glaciations may have been limited by persistent detrital and variable $Fe-OX_{HR}$ loading. Further, microbial recycling of organic-rich P at the primary sediment-water interface was insufficient to generate sediment porewater P saturation to spontaneously trigger vast precipitation of calcium-bearing P minerals across the sequence. This observation limits potential diagenetic interference with primary sediment P content. Production of dissolved sheet silicate bound $PO_4^{3-}$ would have been facilitated by grinding of the bedrock by thawing ice sheets, with the generation of sub-glacial acidity[66]. These subglacial acidic conditions, combined with acidic water produced by the immediate post-Snowball high $CO_2$ atmosphere[67], would have sustained chemical leaching of P from rocks, including apatite minerals[68]. The sheet silicate clay minerals that are expected to more easily bind $PO_4^{3-}$ in more acidic conditions[4–6] transported and liberated bound $PO_4^{3-}$ to seawater following contact with higher marine pH[5,6,10] and salinity[11,17]. The sudden decline in leachable sheet silicate P entering the immediate post-Snowball state, followed by abrupt rise in magnetite P content by at least a factor of 200 compared to Pre-Snowball hints at a potential switch in P sink from clays to seawater. The increase in dissolved seawater P promoted primary productivity and oxygenation, with the resultant recycled biomass P captured together with dissolved inorganic P and preserved by $Fe-ox_{HR}$ minerals generated in the oxidized water column (Fig. 9). Taken together, the data indicate a major switch from marine waters with low dissolved $PO_4^{3-}$ content to an enlarged inventory created by a deglaciating Cryogenian Snowball clay factory.

## Methods
### Sampling and sample preparation
Outcrop sampled lithologies and their locations are described in Table S1. Only a subset of representative rocks were analyzed for this study, but the overall collection was used to establish the placement of

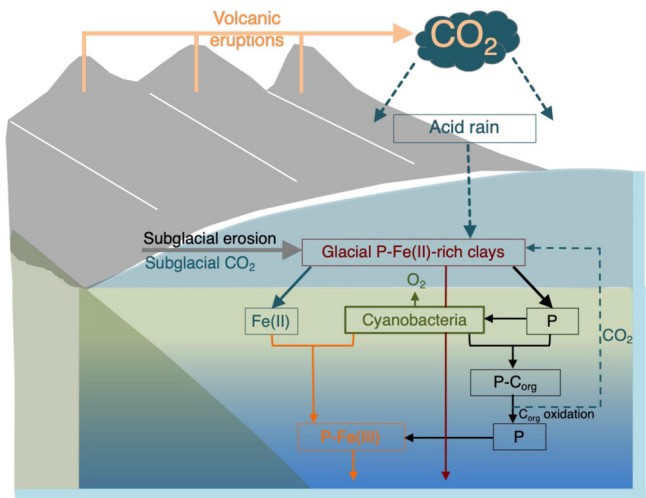

**Fig. 9 | A conceptual model for P cycling across the Sturtian Snowball glaciation.** The model highlights the relationship between glacially generated clays, P and Fe.

samples along sequence stratigraphy. Prior to geochemical analysis, the outer exposed layers of samples were removed and only the unweathered portions were used for analysis. A fraction was milled into a fine powder for geochemical, chemostratigraphic and XRD mineralogical analyses. Slices of whole rock samples were thin sectioned and polished for scanning electron microscopy-energy dispersive spectroscopy (SEM-EDS).

## Mineralogical analysis

X-ray diffraction (XRD) analysis for bulk semi-quantitative qualitative mineralogy analysis was undertaken with a PANalytical Xpert-pro and a Bruker D8 ADVANCE diffractometers at Cardiff University and University of Poitiers, respectively, as previously described[69–71]. SEM-EDS analysis was performed with an Oxford FEI-XL30 fitted Environmental SEM (ESEM) connected to an EDS system at Cardiff University. Analyses were conducted on polished thin sections coated with carbon to a thickness of 14 nm with a BIO-RAD SC500 sputter. The SEM-EDS analyses were run at a working distance of 8.9 mm and a 15 kV accelerating voltage. Backscattered elemental maps, point-specific atomic composition and spectra were combined to identify putative mineral phases. Semi-quantitative XRD analyses of the relative abundance of the mineral assemblages was calculated using the integrated area ratios of the principal peaks after decomposition by the FYTIK program[72].

## Elemental analysis

An approximate 250 mg of bulk rock powders were dissolved using reagent grade acids in closed screw-top Teflon vessels (Savillex) at 90 °C for one day in concentrated 3 ml 40% HF, 3 ml 32% HCl, and 1 ml 65% $HNO_3$. Excess HF was neutralised with 93 ml $H_3BO_3$ (20 g/L) aqueous solution. Elements were measured by inductively coupled plasma atomic emission spectrometry (ICP-AES) using a Horiba Jobin Yvon® Ultima 2 spectrometer with boron as an internal standard. Calibrations were made with ACE, JB2, and WSE international standards, with relative standard deviation of ≤1% for $SiO_2$, ≤2% for the other major elements, and ≤5% for traces elements.

## Iron speciation analysis

Chemical extraction of sediment Fe phases was conducted using the sequential Fe extraction protocol of Poulton and Canfield and Fe measured using the ferrozine method[34–36]. Redox trends were further constrained using the distribution of Fe/Al ratios across the studied

section, where average ratios of ~0.55 ± 0.11 are common for oxic waters, with much greater values for anoxic settings[34–36]. Sediment Al/(Al+Fe+Mn) content was used to demonstrate that hydrothermal contribution[25] of Fe to the sediments was negligible and that observed high Fe/Al ratios cannot be associated with hydrothermal activity. The sequential Fe extraction procedure resulted in the quantification of eight operationally defined Fe phases: (1) Total bulk rock Fe ($Fe_T$) from whole rock dissolved powder using hydrofluoric acid (HF); (2) Poorly reactive sheet silicates Fe or sheet silicates ($Fe_{SS}$) extracted with boiling 12 M HCl; (3) Reducible oxyhydr(oxides), including goethite, akageneite and hematite Fe ($Fe$-oxy) with sodium dithionite, and magnetite Fe with ammonium oxalate ($Fe_{Mag}$); (4) Carbonate-associated Fe ($Fe_{Carb}$) extracted with sodium acetate; (5) pyrite Fe ($Fe_{Py}$) extracted using the chromium reduction method. (6) Unreactive silicate Fe ($Fe_{URS}$) was determined as $Fe_{URS} = (Fe_T) - (Fe_{SS} + Fe\text{-oxy} + Fe_{Mag} + Fe_{Carb} + Fe_{Py})$.

## Chemically extractable Fe-bound P analysis

Co-extracted P associated with the Fe phases was approximated following the optimized Fe-P speciation extraction procedure of Thompson and others[40] for ancient sediments and Fe and P co-measured by ICP-AES. These included Fe carbonate bound-P (Fe-$P_{carb}$), Fe oxyhydr(oxides) associated-P (Fe-ox$_{HR}$-P), magnetite bound-P ($P_{Mag}$) and sheet silicate Fe associated P ($P_{SS}$). The remaining P was assumed tied up mainly with unreactive silicates ($P_{URS}$), first verified by a lack of correlation between total organic carbon and P. We use multiple lines of geochemical evidence, together with Loss on Ignition (LOI) to show that most P trapped in the sediments is bound to inorganic mineral phases. Further, SEM-EDS and XRD analyses were used to show that apatite is not a major mineral phase in the sediments. The potential $P_{URS}$ fraction was then estimated from $P_{URS} = (P_T) - (Fe\text{-}P_{carb} + Fe\text{-ox}_{HR}\text{-}P + P_{mag} + P_{ss})$. Because of the linear correlation between dissolved P and particulate Fe-ox$_{HR}$ in marine sediments[37–39], we further evaluate the distribution of P in the Fe-ox$_{HR}$ phases to approximate the general behaviour of dissolved P across the studied section.

## Stable isotope analysis

For analysis of $\delta^{13}C$ and $\delta^{18}O$ in carbonates, powdered samples weighed into septum vials were flushed with helium, acidified with 99% orthophosphoric acid, and left to react for 24 hours at 60 °C to ensure complete dolomite dissolution. The long-term precision of in-house Carrara marble standard was 0.05‰ for both $\delta^{18}O$ and $\delta^{13}C$ (1 SD). Organic carbon samples were acidified in 10% HCl and left to react for up to two days to remove all inorganic carbon, prior to $\delta^{13}C_{org}$ measurement. Residues were rinsed three times, dried and weighed into tin capsules. The aperture of the autosampler determined the maximum amount of sediment that could be analysed for $C_{org}$ to be 60-80 mg. Because of low $C_{org}$ content, total analysable minimum $C_{org}$ was estimated to be 10 µg. To allow calibration of such small samples, three standards (IAEA-CH6 [$\delta^{13}C = -10.449$], IAEA-600 [$\delta^{13}C = -27.771$] and in-house caffeine [$\delta^{13}C = -33.30$]) were dissolved in de-ionised water to improve homogeneity and allow accurate dosing of small aliquots using a micropipette. Results for IAEA-CH6 and the in-house caffeine were used to estimate a correction function for sample size and size-dependent two-point normalization, which was applied to IAEA-600 as independent standard. The resultant precision was dependent on sample size. The long-term precision for $\delta^{13}C$ was estimated to be 0.09 (1 SD) for routine samples containing ≥100 µg C. However, the standard deviation increases with decreasing sample size to 0.38 for aliquots of 10−30 µg C for the present study (IAEA-600, $n = 15$). Iron isotope analysis was conducted on rock powders as previously described[71].

## Numerical modelling

A four-box ocean $PO_4^{3-}$ model was simulated using the equations described in reference[64]. The model was operated from an initial low oxygen steady state and followed by a pulse of reactive P from a

riverine source or from within the ocean. The latter state represents within the ocean the chemical weathering of glacial debris as a source of $PO_4^{3-}$ to seawater. The initial steady state flux conditions at t = 0 assume half the annual amount of current riverine $PO_4^{3-}$ supply to seawater and increased flux of reductants to the surface ocean equivalent to $1 \times 10^{13}$ mol $O_2$, equating to ~5% PAL of $O_2$ producing a slightly oxic ocean surface and anoxic ocean interior and shelf bottom waters, except for ~20% of the proximal shelf. At t = 0 a pulse of P was delivered to the system by increasing the riverine $PO_4^{3-}$ input by 50% for a total of 5 Myr in duration or adding this additional $PO_4^{3-}$ throughout the ocean interior. In agreement with geological evidence, the model calculates the relationship between $PO_4^{3-}$ and ocean-atmosphere oxygenation and $PO_4^{3-}$ burial rates as authigenic calcium phosphate, $Fe\text{-}ox_{HR}$ and organic matter bound P (refs. [64],[65]).

## Data availability
All data are available in the supplementary information and supplementary Data 1-6.

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

## Acknowledgements

Thanks to Diana Carlsson for help with the fieldwork and sample collection, Stefan Lalonde with trace and major element analyses and Olivier Rouxel for support with iron isotope analysis. We are grateful to Ian Fairchild for inspiring important discussions on the Cryogenian sedimentology and stratigraphy of Islay and the Garvellachs Islands. We thank Flavia Boscolo Galazzo for insightful comments. This work was supported by funds from The European Research Council (ERC) Seventh Framework (F7) program, grant No: 336092. Support is acknowledged from La Région Nouvelle Aquitaine, France.

## Author contributions

E.C.F. designed research; E.C.F., C.B., A.O., A.N., O.B., J.A.B. and J.A. conducted research; L.A .and B.M. developed the numerical model; D.W. and A.S. produced field and stratigraphic maps. E.C.F. and L.L. collected samples; E.C.F., A.E., K.K., O.B., W.F., J.A.B., J.A. and T.L. analysed data. E.C.F. wrote the paper with significant contributions from all co-authors.

## Competing interests

The authors declare no competing interests.
