## [Peer Review File · Nature Communications]

Transient fertilization of a post-Sturtian Snowball ocean margin with dissolved phosphate by clay mineralsReviewer #1 (Remarks to the Author):

The manuscript entitled "Transient fertilization of a post-Sturtian Snowball ocean margin with dissolved phosphate by clay minerals" by Fru et al. studied glaciation deposits of the Dalradian Supergroup. The authors performed sequential extraction, XRD, SEM, and bulk geochemical analysis on carbonate and tillite samples. By analyzing the data, the authors argued that post-glaciation delivered clay minerals to the ocean which fertilized the ocean and afterwards the precipitation of Fe(II) from ocean water column sequestered marine dissolved P in the sediments.

I think the authors properly interpreted the data and the conclusion is straightforward. The normalization of trace elements over Al or Ti to interpret detrital source is widely applied to study trace elemental enrichment. Also, the separation of different mineral phases through sequential extraction approach provides a powerful tool to study the distribution of P among different reservoirs. And the iron speciation analysis which is commonly used in studying ocean anoxia is applied to study the redox environment. Through analyzing the geochemical data, the authors linked two important conclusions on P cycling from previous experimental studies: clay shuttling P and the co-precipitation of Fe and P.

Overall, the experimental data support their argument, and their conclusion is interesting. Thus I suggestion acceptance after minor revision. I only have a couple of comments on the presentation of data. Firstly, I would think scattered plots instead of shadowed figures are better to present the results. Also, a correlation matrix analysis could strength their arguments.

Specific comments:

1. Figure 2: maybe delete the sub-figure nomenclature (e.g., a, b, c) because these alphabet letters are more notable than useful information. Also, in the main text, it's not necessary to use "Fig. 2a", just "Fig. 2" should be enough. Same as figure 5.
2. Figure 2,4,5: maybe plot as scatter instead of shadow might be better?
3. The authors discussed a lot about the correlation between P, Fe, Ti, Al as well as other trace elements in the "bulk geochemistry section". I agree with the discussion and figures 2 and 3 show obvious trends. Actually, talking about correlation, the easiest way is to do a correlation matrix analysis among all the studied proxies. In which case "positive correlation" or "no correlation" are more distinguishable and easier to compare.
4. Figure 4e is the ratio of P over Fe in magnetite phase, and the unit should be 1 instead of ppb.
5. Fig. S1: I see no data in "Fig.S1a"- "Fig.S1d".
6. Line 273: I guess you mean "whether P is associated with Fe"?
7. Line 279-280: should be "The ratio of Fe-oxHR over total Fe content suggests....."?

Reviewer #2 (Remarks to the Author):

Dominic Papineau

Reviewer #2 Attachment on the following page

Review of Chi Fru et al. 2023 – Nat Comm

This manuscript describes new geochemical data from a Sturtian age sequence from Scotland spanning a glaciogenic tillite. The geochemical data (elements, and Fe and P speciation) was collected stratigraphically above, below and within the tillite. This was used to discuss how phosphate was likely bound onto clays, as opposed to Fe-oxide, and delivered as such by rivers. Some modelling is also performed. The conclusion is that phosphate was delivered to seawater during glacial runoff. In my opinion this shows some level of novelty, although the core of this idea (of post-glacial riverine delivery of phosphate to stimulate primary productivity) has been broadly accepted in the community for the past 15 years or so for Proterozoic glacial periods. The manuscript present lots of new data, especially in the supplementary information, some of which is overlooked. As a result, major modifications are needed to incorporate discussion of this new information. This assessment does not question the quality of the work and data, nor the observed correlations between clays and phosphate, but is meant to encourages the authors to give a more accurate picture of what is preserved in these interesting rocks.

The conclusion needs to carefully consider biological phosphate as a source of the enhanced P signal. In fact, the SEM-EDS maps (Fig. S2) of thin sections are all at low spatial resolution (not “high resolution”, as claimed on line 233) so they may not accurately show the presence of small size (<10 micron) apatite crystals, expected from such fine-grained sedimentary rocks. Then, numerous apatite occurrences (which I found hard to find as presented, but they are definitely there – please use some annotations to highlight them) are associated with calcite. This observation contradicts the statement on line 198, which also did not consider the possibilities of carbonate associated phosphate or organic-bound phosphate (see Dodd et al. (2021) Development of carbonate-associated phosphate (CAP) as a proxy for reconstructing ancient ocean phosphate levels. *Geochim. Cosmochim. Acta* 301, 48-69, DOI: 10.1016/j.gca.2021.02.038), neither of which were analysed nor considered in this work. In fact, apatite associated with carbonate in such rocks is rather indicative of diagenetically oxidised biomass. Notably, this is also consistent with the low and negative $\delta^{13}\text{C}_{\text{carb}}$ values reported in Fig. S1e (where the chemostratigraphies are erroneously missing in panels b, c, and d), separately indicative of oxidised biomass. These observations of figures S1 and S2 (which should rather be in the main text, and with a mineral-consistent colour-legend) clearly indicate that the origin of apatite was, at least in part (for the most part?), biological. However, this is not discussed at all in the manuscript, unfortunately. Potential counter-arguments to this comment citing the large size of some apatite (e.g. in Fig. S2d) as possibly indicative of a detrital origin need to carefully consider the impact of metamorphism on these rocks, which is known to lead to larger apatite grains size (see Nutman, 2007: Apatite recrystallisation during prograde metamorphism, Cooma, southeast Australia: implications for using an apatite-graphite association as a tracer in ancient metasedimentary rocks. *Aust. J. Earth Sci.* 54, 981-990. DOI: 10.1080/08120090701488321). In fact, it is likely that these large apatite grains in the tillite are responsible for the higher P concentration observed in the tillite. Hence, the described story is not fully considering the new dataset.

REPLY TO REVIEWER COMMENTS

Reviewer #1 (Remarks to the Author):

The manuscript entitled "Transient fertilization of a post-Sturtian Snowball ocean margin with dissolved phosphate by clay minerals" by Fru et al. studied glaciation deposits of the Dalradian Supergroup. The authors performed sequential extraction, XRD, SEM, and bulk geochemical analysis on carbonate and tillite samples. By analyzing the data, the authors argued that post-glaciation delivered clay minerals to the ocean which fertilized the ocean and afterwards the precipitation of Fe(II) from ocean water column sequestered marine dissolved P in the sediments.

I think the authors properly interpreted the data and the conclusion is straightforward. The normalization of trace elements over Al or Ti to interpret detrital source is widely applied to study trace elemental enrichment. Also, the separation of different mineral phases through sequential extraction approach provides a powerful tool to study the distribution of P among different reservoirs. And the iron speciation analysis which is commonly used in studying ocean anoxia is applied to study the redox environment. Through analyzing the geochemical data, the authors linked two important conclusions on P cycling from previous experimental studies: clay shuttling P and the co-precipitation of Fe and P.

Overall, the experimental data support their argument, and their conclusion is interesting. Thus I suggestion acceptance after minor revision. I only have a couple of comments on the presentation of data. Firstly, I would think scattered plots instead of shadowed figures are better to present the results. Also, a correlation matrix analysis could strength their arguments.

Response

We want to extend our sincere appreciation to the reviewer and for their helpful suggestions. We have undertaken the corrections as suggested, and below we provide explanations to the specific comments.

Specific comments:

1. Figure 2: maybe delete the sub-figure nomenclature (e.g., a, b, c) because these alphabet letters are more notable than useful information. Also, in the main text, it's not necessary to use "Fig. 2a", just "Fig. 2" should be enough. Same as figure 5.

Response

Because the letters are required to identify the individual panels for clear referencing within the text, we have opted to reduce the font size of the letters to allow them to better blend with the figure without creating distraction. Labelling each panel is also in line with Nature Communications' requirement that each panel on a figure must be

labelled, described in the Figure legend, and cited in the text.

2. Figure 2,4,5: maybe plot as scatter instead of shadow might be better?

Response

Plotting the figures as scatter plots give us the same patterns, however the changing dynamics are not properly visualized because of the numerous data points resulting in distracting coloured dots that make the trends extremely difficult to appreciate and trace along the sequence stratigraphy. We also attempted to present the data as box and whisker plots for each time interval, but some of the boxes are too small to enable good display of information for easy comparison. We therefore believe that this is the friendliest approach of visualizing the amount of data incorporated in the graphs, especially for the new Figure 7. Nonetheless, we have worked further on the fonts and font size to enable better display of the new Figure 7.

3. The authors discussed a lot about the correlation between P, Fe, Ti, Al as well as other trace elements in the "bulk geochemistry section". I agree with the discussion and figures 2 and 3 show obvious trends. Actually, talking about correlation, the easiest way is to do a correlation matrix analysis among all the studied proxies. In which case "positive correlation" or "no correlation" are more distinguishable and easier to compare.

Response

The diagrams are drawn to show the subtle geochemical changes for each variable along sequence stratigraphy, which is a standard way of presenting sequence stratigraphy profiling data. Therefore, the correlation referred to are with respect to changes associated with sequence stratigraphy, rather than between the variables. That is, the correlations are structured along sequence stratigraphy, which is critical for explaining time-dependent geochemical fluctuations across the basin. Following the approach suggested, most of these subtle but important geochemical changes would be lost.

For example, to explain Figure 4, we state that *"To test this idea, elemental normalization to Ti – a detrital indicator of authigenic enrichment^{27,28}, shows subtle enrichment consisting of two broad patterns across the stratigraphic profile (Fig. 4j-p). This relationship is highlighted by two observations. First, a reasonable stratigraphic correlation exists between P/Ti and Fe/Ti and to some degree with Mn/Ti ratios (Fig. 3n-p), although generally, MnO does not show significant lithostratigraphic variations (Fig. 4d). Second, a broad minimal correlation is observed for Cr/Ti, V/Ti, Ni/Ti, and Co/Ti (Fig. 4j-m), suggesting that their accumulation was co-regulated by the same sedimentary processes."*

4. Figure 4e is the ratio of P over Fe in magnetite phase, and the unit should be 1 instead of ppb.

Response

ppb has been deleted

5. Fig. S1: I see no data in "Fig.S1a"- "Fig.S1d".

Response

This was a PDF conversion problem. This figure has been redesigned and separated into a new Figure 2 and Figure S1.

6. Line 273: I guess you mean "whether P is associated with Fe"?

Response

Re-written as suggested.

7. Line 279-280: should be "The ratio of Fe-oxHR over total Fe content suggests....."?

Response

Corrected accordingly

Reviewer #2 (Remarks to the Author):

Dominic Papineau

Review of Chi Fru et al. 2023 – Nat Comm

This manuscript describes new geochemical data from a Sturtian age sequence from Scotland spanning a glaciogenic tillite. The geochemical data (elements, and Fe and P speciation) was collected stratigraphically above, below and within the tillite. This was used to discuss how phosphate was likely bound onto clays, as opposed to Fe-oxide, and delivered as such by rivers. Some modelling is also performed. The conclusion is that phosphate was delivered to seawater during glacial runoff. In my opinion this shows some level of novelty, although the core of this idea (of post-glacial riverine delivery of phosphate to stimulate primary productivity) has been broadly accepted in the community for the past 15 years or so for Proterozoic glacial periods. The manuscript present lots of new data, especially in the supplementary information, some of which is overlooked. As a result, major modifications are needed to incorporate discussion of this new information. This assessment does not question the quality of the work and data, nor the observed correlations between clays and phosphate, but is meant to encourage the authors to give a more accurate picture of what is preserved in these interesting rocks.

Response

Thank you very much for the insightful comments. Looking at them, we thought you must be suggesting that the previous Figure S1 and Figure S2 be given greater context in the manuscript. We have therefore separated Figure S1 into two parts. Figure S1a-d is our new

Figure 2. The previous Figure S2 is now our new Figure 3. The latter figure was already explored in the text to the level of novelty it brought. Numerous mineralogical and carbon isotope chemostratigraphic studies have been published for these rocks, and thus we used this knowledge and chemostratigraphy to ascertain the quality of our samples by placement on the sequence stratigraphy and global correlation to the Sturtian glaciation. We compare our observations with published papers (cited in the text) for the same lithostratigraphic sections we have studied. Overall, these data provide mainly supportive information, we therefore took the opportunity to instead explore the core of our work in the limited space. In the end, the new Figure 2 fits nicely with the narrative of the main text. We thank you for the suggestion.

The conclusion needs to carefully consider biological phosphate as a source of the enhanced P signal. In fact, the SEM-EDS maps (Fig. S2) of thin sections are all at low spatial resolution (not “high resolution”, as claimed on line 233) so they may not accurately show the presence of small size (<10 micron) apatite crystals, expected from such fine-grained sedimentary rocks. Then, numerous apatite occurrences (which I found hard to find as presented, but they are definitely there – please use some annotations to highlight them) are associated with calcite. This observation contradicts the statement on line 198, which also did not consider the possibilities of carbonate associated phosphate or organic-bound phosphate (see Dodd et al. (2021) Development of carbonate-associated phosphate (CAP) as a proxy for reconstructing ancient ocean phosphate levels. *Geochim. Cosmochim. Acta* 301, 48-69, DOI: 10.1016/j.gca.2021.02.038), neither of which were analysed nor considered in this work. In fact, apatite associated with carbonate in such rocks is rather indicative of diagenetically oxidised biomass. Notably, this is also consistent with the low and negative $d^{13}C_{carb}$ values reported in Fig. S1e (where the chemostratigraphies are erroneously missing in panels b, c, and d), separately indicative of oxidised biomass. These observations of figures S1 and S2 (which should rather be in the main text, and with a mineral-consistent colour-legend) clearly indicate that the origin of apatite was, at least in part (for the most part?), biological. However, this is not discussed at all in the manuscript, unfortunately. Potential counter-arguments to this comment citing the large size of some apatite (e.g., in Fig. S2d) as possibly indicative of a detrital origin need to carefully consider the impact of metamorphism on these rocks, which is known to lead to larger apatite grains size (see Nutman, 2007: Apatite recrystallisation during prograde metamorphism, Cooma, southeast Australia: implications for using an apatite-graphite association as a tracer in ancient metasedimentary rocks. *Aust. J. Earth Sci.* 54, 981-990. DOI: 10.1080/08120090701488321). In fact, it is likely that these large apatite grains in the tillite are responsible for the higher P concentration observed in the tillite. Hence, the described story is not fully considering the new dataset.

Response

We have corrected the text to avoid the ambiguity brought forward by referring to the presented SEM-EDS images as being of high resolution. They are at the level of the spatial detail they are meant to show and they do show the presence of sparse apatite grains in the post-Snowball Earth samples (see Figure 3). Our focus was to present a picture of dominant minerals that would contribute a strong influence on sediment composition. Apatite scarcity relative to the other dominant minerals like calcite, dolomite, quartz, etc, indicates that it is indeed rare across the entire sequence, with glimpses in the post-Snowball samples. A

straightforward explanation relies on the well-established fact that the formation of diagenetic apatite requires supersaturation of dissolved P in sediment pore waters to trigger vast scale precipitation as seen for phosphorites. It is worth noting that our P concentrations are not up to 18 wt.% expected for phosphorites. If not, apatite deposition occurs only sparingly, as laid out in the text quoted from the manuscript below.

The generalized apatite scarcity, revealed by SEM-EDS mineralogical mapping, imaging and XRD analysis (Fig. S2 & Table S2), suggests that bulk P may be associated with non-calcium bearing minerals such as Fe-ox_{HR} and silicates. It is possible that prevailing environmental conditions limited microbial P transformation of P-rich C_{org} and P-rich Fe-ox_{HR} minerals to trigger diagenetic sediment porewater P supersaturation³⁰. First, dominant detrital P supply would have starved primary sediments of dissolved P. Moreover, with the exception of some immediate post-Snowball samples, across-sequence CaO and P₂O₅ inverse correlations, being up to 79% for the tillites (Fig S5a-b), point to potential P preservation in non-calcium bearing mineral phases. As discussed above, this view is supported by low LOI, low-carbonate facies associated with bulk high P₂O₅/Fe₂O₃ ratios and high LOI, high-carbonate lithologies with bulk low-P₂O₅/Fe₂O₃ ratios (Fig. S4). Overall decreasing P₂O₅ concentration coincides with C_{org} content that increases from post-Snowball to pre-Snowball facies (Fig. S5c), signalling either possible diagenetic P enrichment or loss in primary sediments through microbial oxidation of organic-rich P biomass originating from the water column. Indeed, our lowest δ¹³C_{org} and C_{org} values in the tillites and immediate post-Snowball interval (Fig. 2a-b) coincide with P₂O₅ enrichment compared to the low P₂O₅ pre-Snowball samples (Fig. S5c-d). Based on these observations, microbial recycling of organic-rich P prevailed at the sediment-water interface, but apatite scarcity suggests that this process was insufficient to trigger the simultaneous P supersaturation required to spontaneously precipitate apatite precursor phases and significant substitution in carbonate minerals^{30,31}. As demonstrated below, this relationship was perhaps influenced by abundant Fe-OX_{HR} particles acting as efficient P scavengers at the sediment seawater interface.

A unique observations brought forward by this study is that the comparable high levels of bulk P with fine-grained Cryogenian rock records, suggest they might not always translate directly to high dissolved P bioavailability. This is further important to the consideration that saturated levels of bioavailable P are required to trigger massive apatite formation. Instead, our analysis reveals that such bulk high P concentrations do not always translate to high dissolved P content. Below we provide the revised conclusion, summarizing these observations as recommended.

Our data reveal bulk P content comparable to previously published concentrations for fine grain siliciclastic Cryogenian facies. These data suggest that Cryogenian continental seawater P bioavailability before, during, and after the Sturtian glaciations may have been limited by persistent detrital and variable Fe-OX_{HR} loading. Further, microbial recycling of organic-rich P at the primary sediment-water interface was insufficient to generate sediment porewater P saturation to spontaneously trigger vast precipitation of Ca-bearing P minerals across the sequence. This observation limits potential diagenetic interference with primary sediment P content. Production of dissolved sheet silicate bound PO₄³⁻ would have been facilitated by grinding of the bedrock by thawing ice sheets, with the generation of sub-glacial acidity⁷¹. These

subglacial acidic conditions, combined with acidic water produced by the immediate post-Snowball high CO₂ atmosphere⁷², would have sustained chemical leaching of P from rocks, including apatite minerals⁷³. The sheet silicate clay minerals that are expected to more easily bind PO₄³⁻ in more acidic conditions⁴⁻⁶ transported and liberated bound PO₄³⁻ to seawater following contact with higher marine pH^{5,6,10} and salinity^{11,17}. The sudden decline in leachable sheet silicate P entering the immediate post-Snowball state, followed by abrupt rise in magnetite P content by at least a factor of 200 compared to Pre-Snowball hints at a potential switch in P sink from clays to seawater. The increase in dissolved seawater P promoted primary productivity and oxygenation, with the resultant recycled biomass P captured together with dissolved inorganic P and preserved by Fe-ox_{HR} minerals generated in the oxidized water column (Fig. 9). Taken together, the data indicate a major switch from marine waters with low dissolved PO₄³⁻ content to an enlarged inventory created by a Cryogenian Snowball clay factory.

Reviewer #2 (Remarks to the Author):

This manuscript has moved in the right direction, but still has several shortcomings. In particular:

- I have the impression that several comments from both the other reviewer and me were not all addressed properly. The argument raised in response to my comment on a possible biological source of P, based primarily on the low TOC of these rocks, is still not considering the likely possibility that a massive amount of biomass was oxidised during post-glacial thawing, which is a likely interpretation for the observed negative $\delta^{13}\text{C}_{\text{carb}}$ values of the Bonahaven carbonate. In fact, this observation is almost only used to suggest broad, nameless correlations to other tillite-carbonate bearing sequences of Sturtian age.
- These correlations are also not demonstrated however, also because the age constraints in the stratigraphy of figure 1 are not shown (only an interpretative age scale is shown instead), and no specific comparison is made to at least one other similar Sturtian chemostratigraphic sequence known.
- The point raised by the other reviewer about the absence of the measured data points in most of the figures, and to which I agree fully, shows that both reviewers want to see the measured real data points on the figure panel. It is OK to use coloured shaded areas in your chemostratigraphies, however, you still need to distinctly show ALL your data in these chemostratigraphies and histograms

Here are some other in-text specific problems to address and remediate:

- line 76: 'In this regard...' does not follow; The first statement should not be what is written and rather should follow with one that is more specific to comparable environments, uniformitarian, and to be returned to for later, broader comparisons, so that this should help to setup the discussion. Currently the statement brings in the GOE out of the blue and without much comparisons to be made anyways, here or later.
- Line 127: equation there needs to have the denominator in brackets for mathematical correctness.
- Line 131: it is unclear if you can really talk about a Corg excursion, as you do not report organic-rich rocks and also whether the $\delta^{13}\text{C}_{\text{carbonate}}$ excursion is the postglacial positive or negative that is referred to, because both exist in your (un-shown) data. And what rocks are included in 'immediate post-snowball values' (line) 132?
- Line 150: here and elsewhere in the text, the expression 'tectosilicate plagioclase feldspar mineral' is either over-worded or no point is really ever made about the fact that they are indeed 'tectosilicate' and 'mineral'.
- Line 158: why are there ';' and not simply ','?
- Line 170: yes but kaolinite would also react with alkaline fluids during metasomatism.
- Line 173: vague, your rocks are not demonstrably below the greenschists facies, especially with these phyllosilicates (chlorite, illite, and muscovite).
- Line 174: 'primary features' is too vague.
- Line 332: the wording 'staggered distribution' and appearance in the data is unclear to me.
- Lines 345-347: confusing sentence.
- Line 500: then OK, perhaps your explanation is correct, however there isn't much specific that is discussed about other published concentrations of P content and comments on the relevance of the comparison.
- Line 507: How does it limit this explanation involving diagenesis? Diagenesis is most likely included in this sequence of events.
- Line 513: The only clay mineral mentioned is illite and that is not unambiguously detected, as the XRD data show planes that overlap those of muscovite, no polarised light microscopy images (PPL and CP) at high magnification is shown to confirm it is also not sericite, which could also indicate higher metamorphic grades than presumed or later fluid infiltrations that hydrolysed feldspars.
- Line 523: I think you mean 'deglaciating' Cryogenian snowball.
- Line 563-575: this is all text to be discussed in the discussion section, not in the methods section. It does not make sense to leave out important topics of discussion.
- Supplementary information section needs much better organisation if it is to be truly useful and the author have to revise it to make it acceptable for publication. First this section needs to be

written in the portrait orientation, and then more sub-titles are needed, and more introductory text is needed to more clearly make sense of what this concerns exactly. Only one section currently stands and is supposedly on 'sedimentary rates and iron-based redox proxy', but I see many topics that are not all well connected. It is not clear if these four points represent constraints for the model?

- Table S2: I don't understand the logic behind writing 'semi-quantitative' in the title because your table is very quantitative indeed. What you mean is low precision or low accuracy data or both.

- Figure 1: indicate more precisely where the dolomite concretions occur, please. Add specific age constraints in the stratigraphic column and not only the interpreted age scale. Also, please list the key sedimentological markers in the Bonahaven dolomite (especially what makes it 'evaporitic' as claimed in the main text).

- Figure 2: EDS maps should be accompanied by area scans to show smaller apatite grains, and also possibly with P + Fe and P + Al maps, to better support the arguments in the main text. This is also needed for clarity and to give a better purpose to this figure, which is now in the main text.

- Figures 3, 4, 7, 8, S4, S5: please add data for a more honest and transparent presentation.

- Table S3: some sense of the data within its sampled sequence would be useful as an additional row and to give a better organisation and sense to this table.

- Table S4: In my book, this table suffers from the absence of measurements on standards, controls, duplicates, etc.

So great story overall, but the presentation needs to be updated with all new data explicitly shown in figures, provide high-resolution images of the four specimens to show the absence or presence of smaller apatite crystals in figure 2. The rest of the supplementary information is adequate and useful.

REVIEWER COMMENTS

Reviewer #2 (Remarks to the Author):

This manuscript has moved in the right direction, but still has several shortcomings. In particular:

- I have the impression that several comments from both the other reviewer and me were not all addressed properly. The argument raised in response to my comment on a possible biological source of P, based primarily to the low TOC of these rocks, is still not considering the likely possibility that a massive amount of biomass was oxidised during post-glacial thawing, which is a likely interpretation for the observed negative $\delta^{13}\text{C}_{\text{carb}}$ values of the Bonahaven carbonate. In fact, this observation is almost only used to suggest broad, nameless correlations to other tillite-carbonate bearing sequences of Sturtian age.

Response: If we have given the impression that some of your comments were not fully explained, it was unintentionally and often, we have avoided running with speculations if the evidence is not obvious in our dataset. We would like to indicate that we use our carbonate carbon systematics (not organic carbon as suggested), for chemostratigraphic placement of our samples along the studied sequence. We do this by comparing our data with published observations for these facies (see for example Brasiers and Shields (2000) and Fairchild et al. (2018)). This focus is important to establish the authenticity of our samples, which together with geochronology, is the basis on which these rocks have been broadly correlated chemostratigraphically to Sturtian successions globally. Following your recommendation, we have revised this statement to clarify this point.

We then subsequently discuss the C_{org} systematics with respect to biological processes involved in the recycling of organic matter-rich P at the water-sediment interface and associated apatite and carbonate P sedimentation. We however avoid drawing comparisons with other Sturtian glaciation rocks as such data linking C_{org} systematics to P cycling, particularly apatite, across Sturtian Snowball deposits is limited to enable unambiguous generalization or the naming of specific lithologies elsewhere. We think it is important to recognize novelty in our work. In doing so, we have further highlighted the idea that biological C_{org} oxidation likely contributed to sedimentary P dynamics in the text, consistent with our conceptual model in Figure 9. Finally, as requested, we have provided all the C systematic (C_{org} and carbonates) data in a new Figure S1 as requested. In this figure we have plotted the available data, including C_{org} and carbonate concentrations and C and O isotope variations along sequence stratigraphy. We would like to indicate that we have avoided making strong quantitative speculations on the amount of C_{org} that was potentially present or oxidized in the various sections, since we simply cannot make such an inference from the available data with any degree of certainty. We agree that the negative $\delta^{13}\text{C}$ trends present in all the lithologies show that C_{org} was oxidized across the entire section, but they do not tell us how much C_{org} was present in the water column and sediments or oxidized. Nonetheless, our data are clear

that P abundance shows a strong co-relationship between sheet silicates (potential clays) and iron oxide fluctuations across the studied sequence, the focus of our paper and that these relationships are directly related to aqueous dissolved P content and oxygenation dynamics.

- These correlations are also not demonstrated however, also because the age constraints in the stratigraphy of figure 1 are not shown (only an interpretative age scale is shown instead), and no specific comparison is made to at least one other similar Sturtian chemostratigraphic sequence known.

Response: As explained above, we have plotted the C systematic data along the sequence stratigraphy, together with sample names, the sample coordinates provided in Table S1. The age constraints for the studied sections have been established by previous studies on lithologies that bracket the sampled interval (see for example Brasiers and Shield (2000), Fairchild et al. (2018); Prave et al. (2009); and Sawaki et al. (2010) and references therein. We would like to emphasize that the main placement of our samples along the succession is consistent with established chemostratigraphic variations and lithological correlations that show that the distinct Port Askaig Tillite Formation is a marker of the global Sturtian Cryogenian glaciation and the post-glacial carbonates in the Bonahaven Dolomite Formation the end of the glaciation. Our comparisons are based on these defined lithological types known to represent the distinct pre-glacial, glacial and post-glacial successions. This approach works well for our study, as evidenced by the studies cited above and others, since the point was to study chemostratigraphic changes across these distinct climatic intervals, with respect to P cycling and oxygenation. As it is almost impossible to require age constraints for every single point analyzed along an established sedimentary succession, our approach is consistent with accepted approaches used by the geochemistry community.

- The point raised by the other reviewer about the absence of the measured data points in most of the figures, and to which I agree fully, shows that both reviewers want to see the measured real data points on the figure panel. It is OK to use coloured shaded areas in your chemostratigraphies, however, you still need to distinctly show ALL your data in these chemostratigraphies and histograms.

Response: The data points have been included in all figures.

Here are some other in-text specific problems to address and remediate:

- line 76: 'In this regard...' does not follow; The first statement should not be what is written and rather should follow with one that is more specific to comparable environments, uniformitarian, and to be returned to for later, broader comparisons, so that this should help to setup the discussion. Currently the statement brings in the GOE out of the blue and without much comparisons to be made anyways, here or later.

- Line 127: equation there needs to have the denominator in brackets for mathematical correctness.

Response: The sentence has been rephrased taking this recommendation into consideration. See text.

- Line 131: it is unclear if you can really talk about a C_{org} excursion, as you do not report organic -rich rocks and also whether the δ¹³C_{carbonate} excursion is the postglacial positive or negative that is referred to, because both exist in your (un-shown) data. And what rocks are included in 'immediate post-snowball values' (line) 132?

Response: We have changed excursion to C_{org} trends to remove ambiguity.

- Line 150: here and elsewhere in the text, the expression 'tectosilicate plagioclase feldspar mineral' is either over-worded or no point is really ever made about the fact that they are indeed 'tectosilicate' and 'mineral'.

Response: We have revised this to tectosilicate mineral and have highlighted the fact that their presence indicates an association with the erosion of crystalline bedrocks (see for example Brasiers and Shields, 2000).

- Line 158: why are there ';' and not simply ','?

Response: Replaced as suggested.

- Line 170: yes but kaolinite would also react with alkaline fluids during metasomatism.

Response: We have included this statement in the paragraph.

- Line 173: vague, your rocks are not demonstrably below the greenschists facies, especially with these phyllosilicates (chlorite, illite, and muscovite).

Response: We agree with this observation, which is why we made the statement that *Although chlorite can also form as a low-grade metamorphic mineral, with further transformation to muscovite at higher temperatures, the mineralogy of the facies is consistent with their suggested low metamorphic grade and observed preservation of primary features*¹⁸⁻²³.

We would very much like to emphasize that these rocks show no evidence for high-grade metamorphic alteration. These observations are evident from our mineral composition and with conclusions in the cited papers describing the studied succession and geological setting.

- Line 174: 'primary features' is too vague.

Response: We are unsure what is required here as we are simply suggesting that primary features of these rocks are preserved as opposed to secondary features resulting from post-depositional alteration, based on the cited published observations.

- Line 332: the wording 'staggered distribution' and appearance in the data is unclear to me.

Response: Deleted.

- Lines 345-347: confusing sentence.

Response: Rephrased to: *We assume that the rise of appreciable P enrichment in immediate post-Snowball magnetite grains relative to the tillites, points to increasing immediate post-Snowball co-precipitation of magnetite and dissolved PO_4^{3-} out of seawater.*

- Line 500: then OK, perhaps your explanation is correct, however there isn't much specific that is discussed about other published concentrations of P content and comments on the relevance of the comparison.

Response: This observation is shown in Figure 6, where our samples are superimposed on Earth's sedimentary P record across Earth history and cited in the discussion.

- Line 507: How does it limit this explanation involving diagenesis? Diagenesis is most likely included in this sequence of events.

Response: Our statement doesn't rule out diagenetic formation of apatite via the oxidation of P-rich C_{org} . However, the fact that apatite only becomes relatively pronounced in the immediate post-Snowball facies, suggests that diagenetic enrichment or loss of P through this process was likely limited across most of the sampled sequence.

- Line 513: The only clay mineral mentioned is illite and that is not unambiguously detected, as the XRD data show planes that overlap those of muscovite, no polarised light microscopy images (PPL and CP) at high magnification is shown to confirm it is also not sericite, which could also indicate higher metamorphic grades than presumed or later fluid infiltrations that hydrolysed feldspars.

Response: There is no sericite in these samples. We have not observed sericite-like phases by SEM-EDS analysis, consistent with our XRD analysis. We emphasize again that these samples have not experienced high-grade metamorphism as established by our data and supported by the cited literature. The mineralogical analysis was designed for describing the quality of our samples across the distinct climatic zones and was deemed sufficient for the purpose of this study. This is because the core of our work relies on the exhaustive chemical extraction experiments that link sheet silicate P and iron oxides P phases to oxygenation. The idea of a combination of high resolution imaging techniques is the subject of another study which we have completed for a paper currently written for

Precambrian Research which we would be very much happy to share if this reviewer wants to see the data. The scale of the analysis includes detailed characterization of the independent mineral grains and their associations with P across the various lithologies linked to authigenic and detrital mineral phases, the unnecessary details which would dilute the focus of the present paper. Based on the results, we can say with great certainty that these rocks show no obvious signs of high post-depositional compaction and presence of high-grade metamorphic minerals and the transformation of primary important clay minerals.

- Line 523: I think you mean 'deglaciating' Cryogenian snowball.

Response: Revised as suggested.

- Line 563-575: this is all text to be discussed in the discussion section, not in the methods section. It does not make sense to leave out important topics of discussion.

Response: Moved into the main text where redox implications are discussed.

- Supplementary information section needs much better organisation if it is to be truly useful and the author have to revise it to make it acceptable for publication. First this section needs to be written in the portrait orientation, and then more sub-titles are needed, and more introductory text is needed to more clearly make sense of what this concerns exactly. Only one section currently stands and is supposedly on 'sedimentary rates and iron-based redox proxy', but I see many topics that are not all well connected. It is not clear if these four points represent constraints for the model?

Response: Figures have been improved and data points inserted as suggested. The supplementary section is divided into three distinct sections, i.e., supplementary information where we discuss the sedimentation rates, Supplementary Figures and Supplementary sections that are accompanied by descriptive legends. With respect to redox, sedimentary rates are important because fast rates can dilute Fe content, especially with respect to rapid detrital overloading. We do this by comparing known sedimentation rates for co-eval strata in Svalbard and of known thickness with the thickness of the sampled lithologies, but of unknown sedimentary rate.

- Table S2: I don't understand the logic behind writing 'semi-quantitative' in the title because your table is very quantitative indeed. What you mean is low precision or low accuracy data or both.

Response: Semi-quantitative mineralogical analysis is a widely used standard procedure where XRD data are used to estimate the relative concentration of mineral fractions estimated from the integrated intensities of XRD spectral peaks. These data give a realistic representation of the relative content of various mineral fractions present in the samples. This semi-quantitative analysis is done using the software described in *Wojdyr, M. 2010*.

Fityk: a general-purpose peak fitting program. Journal of applied crystallography 43(5), pp. 1126–1128. See examples of application below.

- 1) *Gisèle Lecomte-Nana, Jean-Pierre Bonnet, Nibambin Soro. Influence of iron on the occurrence of primary mullite in kaolin based materials: A semi-quantitative X-ray diffraction study. Journal of the European Ceramic Society, 113, 669-677.*
- 2) *Talero R. et al., 2011. Comparative and semi-quantitative XRD analysis of Friedel's salt originating from pozzolan and Portland cement. Construction and Building Materials 25, 2370-2380.*
- 3) *Schoen R. 1962. Semi-Quantitative analysis of chlorites by X-Ray diffraction. American Mineralogist 47 (11-12), 1384–1392.*

- Figure 1: indicate more precisely where the dolomite concretions occur, please. Add specific age constraints in the stratigraphic column and not only the interpreted age scale. Also, please list the key sedimentological markers in the Bonahaven dolomite (especially what makes it 'evaporitic' as claimed the main text.

Response: We would like to stress that the provided sequence stratigraphy is drawn according to the accepted stratigraphy of this succession in the published literature, showing key transitional boundaries with respect to lithotype. The dolomite concretions mentioned in the disrupted beds in the BDF are a minor component of the Bonahaven Dolomite Formation to make them a prominent feature on the stratigraphic map. Moreover, we have not sampled them. Further, what is important to note is that these samples are mostly dominated by a dolomite cement, especially in the Lossit Limestone Formation and the Bonahaven Dolomite Formation where carbonates can make up >50 wt.% of the sampled facies. This carbonate cement is highlighted by the SEM photomicrographs in Figure 3. There are additional minor sedimentological features which simply cannot be included in the stratigraphy as they do not represent important marker transitional boundaries, for example mud cracks, etc.

Finally, we would like to emphasize that the mineralogy of these rocks, as referenced in the geological setting section, has been studied extensively. Thus, the focus of this study was not to produce information already established by other studies, but to use our mineralogical data to ensure that our sampling strategy is consistent with sequence stratigraphy. For example, we have not made any unvalidated claims about evaporites. This information is reported in the provided references. Neither was our focus on high-resolution mineralogical characterization of these rocks, but on using chemical extraction experiments to investigate the link between sheet silicates (approximated total clay composition) and iron oxide phases to P cycling and oxygenation across the sequence. We believe that we succeeded in establishing this relationship.

- Figure 2: EDS maps should be accompanied by area scans to show smaller apatite grains, and also possibly with P + Fe and P + Al maps, to better support the arguments in the main text. This is also needed for clarity and give a better purpose to this figure, which is now in the main text.

Response: We have provided a new SEM-EDS Figure S9 showing individual P particles and their association with Fe in the glaciogenic tillites and the immediate post-Snowball interval where our samples record co-increase in P enrichment and oxygenation following the transition in the immediate post-Snowball.

- Figures 3, 4, 7, 8, S4, S5: please add data for a more honest and transparent presentation.

Response: Figures have been updated with data points included.

- Table S3: some sense of the data within its sampled sequence would be useful as an additional row and to give a better organisation and sense to this table.

- Response: Requested information included.

- Table S4: In my book, this table suffers from the absence of measurements on standards, controls, duplicates, etc.

Response: In the methods section we provide a reference to the method used for Fe isotope analysis which includes all the points raised above. Our data are generated in the isotope geochemistry laboratory of Institut Français de Recherche pour l'Exploitation de la Mer Brest, with many decades of Fe isotope analysis. The Table displays the data.

So great story overall, but the presentation needs to be updated with all new data explicitly shown in figures, provide high-resolution images of the four specimens to show the absence or presence of smaller apatite crystals in figure 2. The rest of the supplementary information is adequate and useful.

Response: Revised as suggested.